# SpikeCLR: Self-Supervised Contrastive Learning for Visual Representations with Spiking Neural Networks

Chengwei Zhou [1]   Gourav Datta [1]

## Abstract

Spiking Neural Networks (SNNs) offer a promising alternative to traditional artificial neural networks by leveraging sparse, event-driven computation that closely mimics biological neurons. When deployed on neuromorphic hardware, SNNs enable substantial energy savings due to their temporal and asynchronous processing. However, training SNNs remains fundamentally difficult because the non-differentiable nature of spike generation breaks the bidirectional gradient flow required in modern self-supervised learning (SSL) frameworks. In this work, we introduce the first fully SSL framework for SNNs that scales to large-scale visual tasks without requiring labeled fine-tuning. Our method leverages intrinsic spike-time dynamics by aligning representations across time steps and augmented views. To address gradient mismatch during surrogate training, we propose the *MixedLIF* neuron model, which combines a spiking path with an antiderivative-based surrogate path during training to stabilize optimization, while retaining a fully spiking and energy-efficient architecture at inference. We also introduce two temporal objectives, Cross Temporal Loss and Boundary Temporal Loss, that align multi-time-step outputs to improve learning efficiency. Our approach achieves competitive performance across both ResNet- and Vision Transformer-based SNNs on both static and neuromorphic datasets. Our approach further generalizes through transfer learning from ImageNet-1K to downstream tasks. Notably, our self-supervised SNNs match or exceed the performance of some non-spiking SSL models, demonstrating both representational strength and energy efficiency.

## 1. Introduction

Spiking Neural Networks (SNNs) (Maass, 1997) are a class of biologically inspired models that process information through sparse, event-driven spikes (Pfeiffer et al., 2018), rather than continuous-valued activations. This asynchronous, spike-based computation allows SNNs to perform operations only when necessary, leading to significantly lower activity and reduced energy consumption. During inference, SNNs replace costly multiply-and-accumulate operations with simple accumulations, enabling orders-of-magnitude gains in efficiency. These benefits are particularly pronounced when deployed on neuromorphic hardware such as Intel's Loihi (Davies et al., 2021) and SynSense's Speck processors (Richter et al., 2023), which are optimized for low-power, event-driven processing. However, training SNNs remains a significant challenge. The discrete nature of spike generation makes them non-differentiable and incompatible with standard backpropagation. Surrogate gradient methods (Neftci et al., 2019; Wu et al., 2018; Huh & Sejnowski, 2017; Bellec et al., 2018a) have enabled gradient-based training by approximating the spike function with smooth surrogates, leading to advances in supervised SNNs across convolutional and transformer architectures. However, these approaches remain reliant on labeled data and are yet to fully unlock the potential of SNNs in label-scarce settings.

In contrast, self-supervised learning (SSL) has transformed representation learning in ANNs by eliminating the reliance on manual labels and enabling models to extract generalizable features directly from raw data (He et al., 2022; Oquab et al., 2023; Caron et al., 2021; Zbontar et al., 2021). However, these advances have not transferred effectively to the spiking domain. Existing attempts either depend on supervised fine-tuning after pretraining (Qiu et al., 2023; Hagenaars et al., 2021) or adapt ANN-style pretext tasks (Zhou et al., 2024b) without leveraging the distinctive temporal dynamics and sparsity of SNNs. As a result, prior SNN "SSL" methods remain limited to low-resolution benchmarks such as MNIST or CIFAR and do not generalize to high-resolution datasets or dense prediction tasks. Critically, no existing work has demonstrated a *fully self-supervised*, scalable SNN framework capable of operating at the level

---

[1]Department of Electrical, Computer and Systems Engineering, Case Western Reserve University, Cleveland, USA. Correspondence to: Chengwei Zhou <chengwei.zhou@case.edu>, Gourav Datta <gourav.datta@case.edu>.

*Proceedings of the 43rd International Conference on Machine Learning*, Seoul, South Korea. PMLR 306, 2026. Copyright 2026 by the author(s).

of ImageNet-1K or beyond.

This gap is particularly important because SSL addresses a fundamentally different scalability challenge than supervised training. Supervised SNNs rely on large annotated datasets, which are scarce for both neuromorphic sensors and real-world robotic environments. In contrast, SSL enables scalable representation learning directly from abundant unlabeled sensory streams—precisely the data regime where SNNs are most compelling due to their event-driven efficiency and temporal acuity. Solving SSL for SNNs therefore represents a critical step toward building label-efficient, scalable, and biologically grounded spiking models that can operate in realistic, continuously evolving environments.

**Our Contributions**. *We present the first fully self-supervised SNN framework that achieves competitive performance on large-scale pretraining (ImageNet-1K) and successfully transfers to large-scale downstream tasks such as COCO object detection and semantic segmentation, sometimes outperforming non-spiking SSL baselines*, without relying on labeled supervision. Our key insight is to exploit spike timing dynamics as a natural source of temporal diversity, enabling rich representation learning across time. We propose a dual-path contrastive learning framework that integrates a spiking path and an antiderivative-based surrogate path during training, aligning representations across time steps from two augmented views through temporal contrast. Only the spiking path is used at inference, preserving energy efficiency. We further propose two temporal alignment objectives that effectively learn from spike-time dynamics across augmented sequences: *Cross Temporal Loss*, which aligns all time steps, and *Boundary Temporal Loss*, which focuses on the first and final time steps to reduce computational cost. Our method is compatible with both CNN and Vision Transformer (ViT) based SNN architectures and demonstrates strong generalization on both static (ImageNet-1K, CIFAR-10) and neuromorphic (CIFAR10-DVS) datasets as well as strong transfer performance to downstream datasets. Notably, *we show that our self-supervised SNNs can outperform non-spiking SSL models in some settings, highlighting the representational advantage of SNN's temporal dynamics*, while also offering superior energy efficiency during inference.

## 2. Background & Related Work

### 2.1. Spiking Neural Networks

Spiking Neural Networks (SNNs) process information through discrete spike events over time, enabling sparse, event-driven computation that is attractive for energy-efficient learning systems and neuromorphic deployment (e.g., Loihi 2 (Davies et al., 2021)). In the context of this work, the key property of SNNs is their temporal state evo-

lution—an attribute that offers potential advantages for representation learning across multiple augmented views, but also introduces the central technical challenge we address: the spike function is non-differentiable, preventing the bidirectional gradient flow required by modern self-supervised learning (SSL).

The LIF neuron is the standard computational unit for deep SNNs. Its discrete-time dynamics are

$$H_t = \tau V_{t-1} + W X_t, \ S_t = \Theta(H_t - V_{th}),$$
$$V_t = (1 - S_t) \cdot H_t + S_t \cdot V_{reset}, \tag{1}$$

where $H_t$, $V_t$, and $S_t$ denote the integrated current, membrane potential, and spike output at time $t$. These temporal updates are crucial in our SSL setting because they govern how information propagates across multiple time steps and across augmented views.

Despite the non-differentiability of $\Theta(\cdot)$, recent progress in surrogate-gradient (SG) learning has made supervised deep SNNs practical at scale (Neftci et al., 2019). SG methods replace the spike with a smooth surrogate during backpropagation, enabling stable optimization in CNN-based (Fang et al., 2021; Xiao et al., 2022; Meng et al., 2023; Du et al., 2025) and Transformer-based SNNs (Yao et al., 2023; Zhou et al., 2022b; Yao et al., 2024). However, all existing SG-trained SNNs operate strictly in *supervised* regimes, because the surrogate formulation still does not support the *bidirectional, cross-view gradient flow* required by contrastive or redundancy-reduction SSL objectives. This limitation directly motivates our MixedLIF design, which preserves surrogate-based differentiability while enabling reliable gradient exchange between augmented samples. Finally, on the deployment side, neuromorphic platforms such as Loihi 2, together with software stacks like `Lava-DL` (Team, 2023), provide efficient inference backends for SG-trained SNNs, further motivating SSL frameworks that can exploit both event-driven computation and learned temporal structure.

### 2.2. Self-Supervised Learning

SSL has become a dominant strategy in representation learning, especially in vision and language domains, due to its ability to learn from raw, unlabeled data. In the context of artificial neural networks (ANNs), SSL has matured into a powerful alternative to supervised training, achieving competitive performance on benchmarks such as ImageNet (Oquab et al., 2023). These models are often pretrained on large-scale unlabeled datasets and fine-tuned for downstream tasks, making them both scalable and versatile. However, top-performing methods still require billions of images, extensive data augmentations, and prolonged training times, limiting their practicality in edge and low-resource settings. SSL methods in ANNs have progressed from early contrastive objectives to more re-

cent non-contrastive and reconstruction-based approaches. Early frameworks like SimCLR (Chen et al., 2020b) and MoCo (He et al., 2020; Li et al., 2021) used contrastive losses to align representations from augmented views while distinguishing them from other samples. Building on this, Hard Negative Mixing (HNM)(Kalantidis et al., 2020) enhanced contrastive learning by interpolating informative negative samples, and i-Mix(Lee et al., 2021) introduced a domain-agnostic label and feature mixing strategy to improve robustness. This line of work was later followed by non-contrastive methods such as BYOL (Grill et al., 2020b), and Barlow Twins (Zbontar et al., 2021), which eliminated the need for negative samples and focused instead on redundancy reduction or cross-view alignment. Parallel to these advances, masked image modeling has emerged as an effective pretext task. Inspired by BERT in NLP, methods like MAE (He et al., 2022), iBOT (Zhou et al., 2022a), Mask-Feat (Wei et al., 2022), DINO (Caron et al., 2021; Oquab et al., 2023) and MSF (Assran et al., 2022) train models to reconstruct missing image patches or predict intermediate features from occluded inputs.

Despite significant progress in self-supervised learning for ANNs, extending SSL to Spiking Neural Networks (SNNs) remains challenging due to the discontinuous nature of spike-based activations, which complicates direct adoption of conventional SSL techniques (Nandakumar et al., 2020; Zhou et al., 2024a). Although SNNs are biologically well-matched to self-supervision, their temporal dynamics introduce unique optimization barriers. Recent efforts have explored contrastive and masked-image SSL for SNNs (Qiu et al., 2023; Hagenaars et al., 2021; Bahariasl & Kheradpisheh, 2024; Singhal et al., 2024; Zhou et al., 2024b; Lv et al., 2025; Dong et al., 2025). Spikformer V2 (Zhou et al., 2024b) adapts masked image modeling to SNNs and achieves strong accuracy on ImageNet, though its evaluation protocol relies on supervised fine-tuning rather than a fully self-supervised pipeline. SpikeClip (Lv et al., 2025) learns vision-language representations by distilling knowledge from a pretrained CLIP teacher into an SNN student, targeting multimodal alignment rather than unsupervised image representation learning. PredNext (Dong et al., 2025) addresses video understanding through a temporal prediction auxiliary module that can be appended to existing ANN-style SSL objectives, with evaluation focused on video action recognition benchmarks. Despite these advances, existing approaches most rely on supervised fine-tuning, achieve limited accuracy, or are restricted to neuromorphic datasets. Moreover, these approaches largely adapt ANN-based SSL frameworks without explicitly exploiting spike timing and temporal structure. As a result, despite advances in supervised SNN training (Eshraghian et al., 2023), scalable and effective self-supervised learning for SNNs remains largely unexplored.

# 3. Method

In this section, we describe our proposed SSL framework for SNNs. While surrogate gradients mitigate spike discontinuity for supervised objectives, they do not preserve consistent cross-sample gradients required for self-supervised objectives. This motivates *MixedLIF*, a novel neuron module designed to stabilize surrogate-based training in self-supervised regimes. To further exploit the temporal structure of SNNs, we propose two loss functions: *Cross Temporal Loss* and *Boundary Temporal Loss*, which align representations across time steps.

## 3.1. MixedLIF

To develop an SSL framework for SNNs, we draw inspiration from the Barlow Twins (Zbontar et al., 2021) method. As illustrated in Figure 1, we present the *MixedLIF* neuron, a dual-path design where one path uses standard spiking dynamics and the other employs a range-continuous activation. This design enables gradient-based optimization and alleviates the gradient mismatch issue encountered in SNN training. *During inference, only the spiking path is used, preserving the energy efficiency inherent to SNNs.* In particular, our SSL framework comprises two paths, denoted as $A$ and $B$, which process two independently distorted and time-augmented views, $X^A$ and $X^B$, of the same input (see Figure 2). We adopt the same spatial augmentations as Barlow Twins (random crop, flip, color jitter). For event-based datasets, we additionally apply temporal augmentations (time-reversal, frame dropout, and polarity jitter) to exploit the inherent temporal structure of event streams. For static datasets, the input image is simply replicated across all time steps (direct encoding (Rathi et al., 2020)), as there is no meaningful temporal variation to augment. Path $A$ consists of standard LIF neurons. Path $B$, in contrast, uses the antiderivative of the surrogate function used in $A$, producing smooth, non-spiking activations to facilitate gradient flow. Specifically, the SG in path $A$ is defined as

$$SG(H_t^A) = \begin{cases} \frac{1}{\alpha} & , -\frac{\alpha}{2} \leq H_t^A \leq \frac{\alpha}{2} \\ 0 & , \text{ otherwise} \end{cases} \quad (2)$$

where $H_t^A$ is the accumulated input current at time step $t$, and $\alpha$ controls the width of the SG function, governing the steepness of the clipping-differentiable activation function. The antiderivative used in path B, computed by integrating $SG(H[t])$ the SG function over the window $[V_{\text{th}} - \frac{\alpha}{2}, V_{\text{th}} + \frac{\alpha}{2}]$ is

$$ReLU_{\text{clip}}(H_t^B) = \text{clip}\left(\frac{1}{\alpha}(H_t^B - V_{\text{th}}) + \frac{1}{2}, 0, 1\right) \quad (3)$$

where $\text{clip}(x, 0, 1) = \min(\max(x, 0), 1)$ ensures that the output is bounded between 0 and 1. The outputs of the

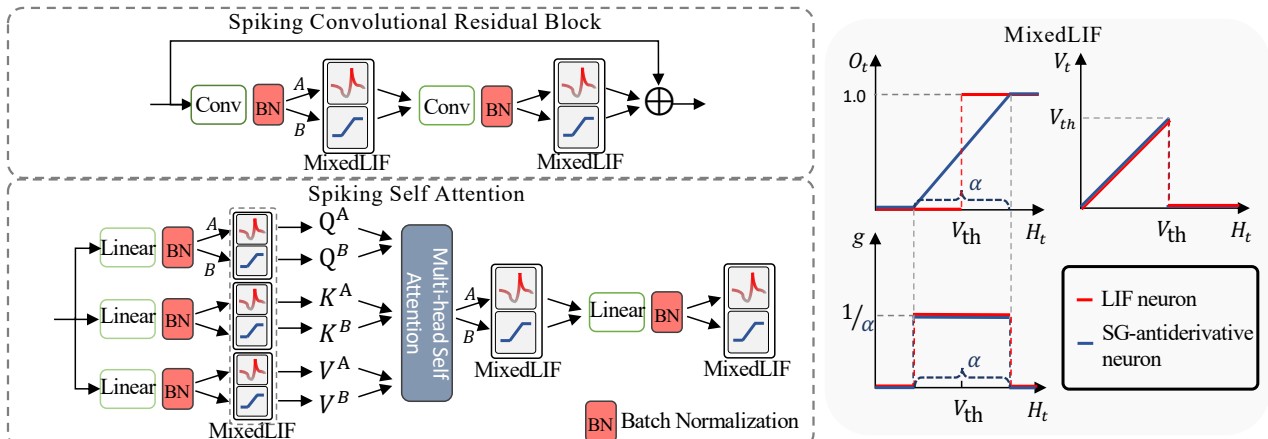

*Figure 1.* **Left:** Core components of our weight-shared dual-path SNN architecture. Both paths share the same trainable parameters but differ in activations via the MixedLIF neuron: Path $A$ emits spikes using a standard LIF neuron, while Path $B$ uses the SG-antiderivative neuron to update using true gradients. **Right:** Visualization of the MixedLIF neuron's dynamics, including input current $H_t$, output spikes $O_t$, post-spike membrane potential $V_t$, and the surrogate gradient used for training.

MixedLIF neuron for paths $A$ and $B$ are then given as

$$O_t^A = \Theta(H_t^A - V_{\text{th}}), \ O_t^B = ReLU_{\text{clip}}(H_t^B). \quad (4)$$

The corresponding membrane potentials are then updated as

$$V_t^A = (1 - O_t^A) \cdot H_t^A + V_{\text{reset}}O_t^A, \quad (5)$$

$$V_t^B = \left(1 - \Theta\left(O_t^B - \frac{1}{2}\right)\right) \cdot H_t^B$$
$$+ V_{\text{reset}}\Theta\left(O_t^B - \frac{1}{2}\right). \quad (6)$$

Here we apply a hard reset for the spiking path and a refined hard reset-like mechanism for path B, setting $V_{\text{reset}}=0$ in both cases. We use hard resets rather than soft resets to avoid residual membrane activity, which could introduce distributional shifts between the two paths. Such shifts, if accumulated across layers, can degrade the effectiveness of the two-path SSL training.

Our framework is compatible with both CNN and ViT backbones (see Fig. 1), which we adapt into their spiking counterparts. We use 32-bit fixed point representation for the weights and membrane potentials. For the convolutional architecture, we follow the ResNet design, replacing ReLU with our MixedLIF neuron. For the ViT backbone, similar to Spikformer, we replace LayerNorm with BatchNorm, substitute GeLU with a spiking activation module (LIF in Spikformer, MixedLIF in our framework), and remove softmax attention. Notably, for Spikformer backbones, we obtain the final representation by averaging output features across the token dimension in the last stage. In contrast, convolutional backbones naturally produce a $1\times1$ spatial feature map due to progressive striding, eliminating the need for additional

averaging. Additionally, we employ a non-spiking projection head following the Barlow Twins design (Zbontar et al., 2021) to enhance representation learning. This component is lightweight relative to the spiking backbone and adds negligible overhead ($<1\%$ in parameters), preserving overall efficiency of the SNN. Converting the projection head to spiking incurs a minor 0.8% accuracy drop, providing a promising direction for achieving fully spiking architectures compatible with neuromorphic hardware.

### 3.2. Loss Functions

SNNs naturally encode information over time through asynchronous spike dynamics, providing an opportunity to leverage rich temporal representations that go beyond static frame-based learning. Traditional SSL methods such as Barlow Twins (Zbontar et al., 2021) operate on static embeddings by aligning representations from two augmented views of the same input. When naively extended to SNNs, this approach would involve aligning embeddings at corresponding time steps, meaning the first time step of path $A$ is matched with the first time step of path $B$, and so on. However, such formulations overlook the temporal structure of spike-based computation, where information is not isolated per time step but distributed and causally linked across time. To fully exploit this temporally entangled structure, we propose the *Cross Temporal Loss*, a fundamentally spiking-aware objective that aligns representations across all pairs of time steps between the two augmented input streams. Rather than treating each time step as independent, our loss captures temporal cross-correlations that reflect how spike dynamics evolve and co-adapt over time (see Figure 2). This leads to feature representations that are not only

robust to augmentations but also sensitive to the intrinsic causal and sequential structure of spikes, enabling the model to better generalize across time-varying inputs.

Formally, let $Z_t^A$ and $Z_{t'}^B$ denote the output embeddings at time steps $t$ and $t'$ from two augmented views $A$ and $B$, respectively. We compute the cross-correlation matrix $\mathcal{C}_{ij}(Z_t^A, Z_{t'}^B)$ between all pairs of time steps $t$ and $t'$ as

$$\mathcal{C}_{ij}(Z_t^A, Z_{t'}^B) = \frac{\sum_b z_{b,t,i}^A\, z_{b,t',j}^B}{\sqrt{\sum_b (z_{b,t,i}^A)^2}\,\sqrt{\sum_b (z_{b,t',j}^B)^2}} \quad (7)$$

where $z_{b,t,i}^A$ represents the scalar output embedding term in the channel dimension $i$ of sample $b$ in time step $t$ in path $A$. The Cross Temporal Loss is then defined as

$$\mathcal{L}_{\mathcal{CT}} = \frac{1}{T} \sum_{t=1}^{T} \sum_{t'=1}^{T} \sum_{\{Z_t^p, Z_{t'}^{p'}\} \in \mathcal{P}(Z),\ Z_t^p \neq Z_{t'}^{p'}}$$
$$\left[ \sum_i \left( 1 - \mathcal{C}_{ii}^2(Z_t^p, Z_{t'}^{p'}) \right) + \lambda \sum_i \sum_{j \neq i} \mathcal{C}_{ij}^2(Z_t^p, Z_{t'}^{p'}) \right] \quad (8)$$

where, $T$ denotes the total number of time steps, and $\lambda$ is a trade-off hyperparameter. The set $\mathcal{P}(Z)$ represents all multisets formed by pairing embeddings across time steps from both augmented views $Z^A$ and $Z^B$, i.e., $\mathcal{P}(Z) = \left\{ \{Z_t^p, Z_{t'}^{p'}\} \,\middle|\, p, p' \in \{A, B\},\ t, t' \in \{1, 2, \ldots, T\} \right\}$. This loss function promotes invariance by aligning the diagonal elements of the cross-correlation matrix to 1 and reduces redundancy by minimizing the off-diagonal elements.

However, the computational complexity of this loss scales quadratically with the number of time steps $O(T^2)$, which may increase the training complexity significantly for long sequences. To mitigate this, we also propose the *Boundary Temporal Loss*, which focuses on aligning representations at the initial ($t=1$) and final ($t=T$) time steps. As shown in Fig. 2, this provides a computationally efficient alternative while still capturing essential temporal dynamics. This boundary loss is formulated as

$$\mathcal{L}_{\mathcal{BT}} = \frac{1}{2} \sum_{t \in \{1, T\}} \sum_{t' \in \{1, T\}} \sum_{\{Z_t^p, Z_{t'}^{p'}\} \in \mathcal{P}(Z),\ Z_t^p \neq Z_{t'}^{p'}}$$
$$\left[ \sum_i \left( 1 - \mathcal{C}_{ii}^2(Z_t^p, Z_{t'}^{p'}) \right) + \lambda \sum_i \sum_{j \neq i} \mathcal{C}_{ij}^2(Z_t^p, Z_{t'}^{p'}) \right] \quad (9)$$

The proposed loss minimizes redundancy at the boundaries while retaining key temporal features.

We adopt BTL as the default temporal loss for static datasets and CTL for event-driven datasets, unless otherwise specified. For static datasets, LIF dynamics produce smooth

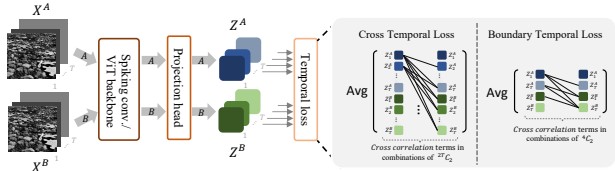

*Figure 2.* Overview of our self-supervised training framework. Two independently distorted and time-augmented views, $X^A$ and $X^B$, are passed through separate processing paths $A$ and $B$. Path $A$ uses Leaky Integrate-and-Fire (LIF) neurons to generate discrete spike outputs $Z^A$, while path $B$ employs the antiderivative of the LIF surrogate function used in path $A$, yielding range-continuous outputs $Z^B$. Both sequences are then projected and compared via Cross Temporal or Boundary Temporal Loss to encourage temporally consistent and invariant representations.

temporal evolution where intermediate states are well approximated by boundary frames, making BTL sufficient to capture the dominant temporal structure; we provide analytical support for this in Appendix A. For event-driven datasets, each timestep receives distinct sparse input, producing non-monotonic trajectories that BTL's boundary-only alignment cannot fully recover. As shown in the BTL/CTL ablation in Appendix C, CTL consistently outperforms BTL across event-driven benchmarks, confirming that exhaustive pairwise temporal alignment is necessary for this data modality.

---

**Algorithm 1** Proposed Dual-Path SSL Training with MixedLIF

**Require:** shared parameters $\theta$, learning rate $\eta$, epochs $N$, batch size $B$, time steps $T$
1: **for** epoch $= 1$ to $N$ **do**
2:     **for** each batch $X$ of size $B$ **do**
3:         $X^A, X^B \leftarrow$ spatial_augment($X$)
4:         apply same temporal augmentation to $X^A, X^B$ over $T$ time steps (only for static datasets)
5:         Initialize $\theta$ using Kaiming normal dist., Initialize membrane potential as $V_0^A=0$ & $V_0^B=0$
6:         **for** $t = 1$ to $T$ **do**
7:             $Z_t^A \leftarrow$ proj_head(backbone($X_t^A, V_{t-1}^A; \theta$))
8:             $Z_t^B \leftarrow$ proj_head(backbone($X_t^B, V_{t-1}^B; \theta$))
9:             Update $V_t^A$ and $V_t^B$ according to Eqs. 5 and 6
10:         **end for**
11:         $\mathcal{L}_{\text{CT}} \leftarrow$ CrossTemporalLoss($Z^A, Z^B$) (Eqs. 7, 8)
12:         $\mathcal{L}_{\text{BT}} \leftarrow$ BoundaryTemporalLoss($Z^A, Z^B$) (Eqs. 7, 9)
13:         Compute $g^A, g^B \leftarrow \nabla_\theta \mathcal{L}_{BT}$ (or $\nabla_\theta \mathcal{L}_{CT}$)
14:         $\theta \leftarrow \theta - \eta(g^A + g^B)$
15:     **end for**
16: **end for**

---

### 3.3. Training Mechanism

Our dual-path SSL framework, illustrated in Algorithm 1, improves the stability of SSL in SNNs by aggregating learning signals from both spiking activations in path A and surrogate-based activations in path B. Note that both paths

*Table 1.* Top-1 and top-5 accuracies under linear evaluation on static datasets (ImageNet-1K, CIFAR-10), and a neuromorphic dataset (CIFAR10-DVS).

| Dataset | Method | Backbone | Time steps | Acc.(%) | |
| --- | --- | --- | --- | --- | --- |
| | | | | Top-1 | Top-5 |
| ImageNet-1K | SimCLR (Chen et al., 2020b) | ResNet50 | - | 69.3 | 89.0 |
| | MoCo v2 (Chen et al., 2020c) | ResNet50 | - | 71.1 | - |
| | Barlow-Twins (Zbontar et al., 2021) | ResNet50 | - | **73.2** | **91.0** |
| | Ours (BTL) | Spiking-ResNet50 | 4 | 69.5 | 89.3 |
| | Ours (BTL) | Spikformer-8-512 | 4 | **70.1** | **89.9** |
| CIFAR-10 | Barlow-Twins[1] (Zbontar et al., 2021) | ResNet34 | - | 84.2 | - |
| | Barlow-Twins[1] (Zbontar et al., 2021) | ViT-4-384 | - | 83.9 | - |
| | Contrastive SSL (Bahariasl & Kheradpisheh, 2024) | Spiking CNN | 30 | 62.2 | - |
| | Ours (BTL) | Spiking-VGG16 | 4 | 81.9 | - |
| | Ours (BTL) | Spiking-ResNet34 | 4 | **85.6** | - |
| | Ours (BTL) | Spikformer-4-384 | 4 | **84.9** | - |
| CIFAR10-DVS | Contrastive SSL(Singhal et al., 2024) (Hybrid inputs+FT) | Spiking-ResNet-18 | 10 | **64.1** | - |
| | Ours (CTL) | Spiking-ResNet18 | 10 | 59.4 | - |
| | Ours (CTL) | Spiking-ResNet34 | 10 | 63.9 | - |
| | Ours (CTL) | Spikformer-4-384 | 10 | 61.2 | - |

*Table 2.* Semi-supervised learning on ImageNet-1K and CIFAR-10 using 1% and 10% labelled training samples (Top-1 and Top-5 accuracy).

| Dataset | Method | Backbone | Time steps | 1% | | 10% | |
| --- | --- | --- | --- | --- | --- | --- | --- |
| | | | | Top-1 | Top-5 | Top-1 | Top-5 |
| ImageNet-1K | SimCLR | ResNet50 | - | 48.3 | 75.5 | 65.6 | 87.8 |
| | Barlow-Twins | ResNet50 | - | **55.0** | **79.2** | **69.7** | **89.3** |
| | Ours | Spiking-ResNet50 | 4 | 52.6 | 77.5 | 67.2 | 88.3 |
| | Ours | Spikformer-8-512 | 4 | 51.7 | 76.4 | 66.6 | 88.1 |
| CIFAR-10 | Barlow-Twins[1] | ResNet34 | - | 73.6 | - | 85.6 | - |
| | Ours | Spiking-VGG16 | 4 | 73.9 | - | 85.9 | - |
| | Ours | Spiking-ResNet34 | 4 | **75.1** | - | **87.6** | - |
| | Ours | Spikformer-4-384 | 4 | 74.5 | - | 87.1 | - |

share the same trainable parameters and process the same input sample in parallel, with each path maintaining its own neuron dynamics. All weights are initialized identically using Kaiming normal distribution (He et al., 2015), ensuring symmetry at the start of training. Despite this, the difference in activation functions yields diverse gradient signals that enhance representation learning. The projected outputs from these paths are then used to compute the SSL objectives defined in Equations 8 and 9. During backpropagation, the gradients from both paths, $g^A$ and $g^B$ are aggregated to update the shared parameters: $\theta \leftarrow \theta - \eta(g^A + g^B)$, where $\eta$ is the learning rate. This gradient fusion mechanism allows updates to be informed by both discrete spike dynamics and continuous surrogates, improving training stability and convergence. Our framework is agnostic to the specific surrogate function; any differentiable or piecewise-smooth approximation can be used. We also explored learning the surrogate slope $\alpha$ and firing threshold $V_{\text{th}}$, but found the gains to be marginal. Consequently, we fix these values and focus on the architectural benefits of our dual-path, shared-weight design.

# 4. Results

In this section, we pretrain our Spiking Neural Networks (SNNs) on ImageNet-1K (Deng et al., 2009), CIFAR-10

(Krizhevsky et al.), and the neuromorphic CIFAR10-DVS dataset (Li et al., 2017), using Spiking-ResNet and Spikformer backbones to validate our approach. For clarity, we note "Spikformer-N-D" indicating the Spikformer is configured with block number of N, and the feature embedding dimension is D. For ImageNet and CIFAR-10 we apply the same spatial augmentations used in SimCLR (Chen et al., 2020b), BYOL (Grill et al., 2020a) and Barlow Twins (Zbontar et al., 2021): random crop and resize with horizontal flip, color distortion and Gaussian blur. For CIFAR10-DVS, we apply temporal augmentation, including time-reversal, frame dropout, and polarity jitter. All experiments were conducted on 8 NVIDIA H200 GPUs. Additional pre-training, fine-tuning, and transfer learning details are provided in Appendix B.

## 4.1. Linear Evaluation

We assess the quality of the learned representations by training a linear classifier on top of frozen features extracted from our dual-path spiking SSL model, trained with the Boundary Temporal Loss for its efficiency. This follows the standard linear evaluation protocol (Zhang et al., 2016; Oord et al., 2018; Bachman et al., 2019), as adopted in (Chen et al., 2020a). As shown in Table 1, our method achieves competitive performance on ImageNet, matching Barlow Twins

on ResNet-50 with Spiking-ResNet50 (73.2% vs. 69.5%), and reaching 70.1% with Spikformer-8-512. For CIFAR-10, Spiking-ResNet34 and Spikformer-4-384 attain 85.6% and 84.9%, outperforming the non-spiking Barlow Twins models (84.2% and 83.9%, respectively). For comparison on the CIFAR10-DVS classification task, (Singhal et al., 2024) reports 64.1% using a Spiking-ResNet18, but their method requires hybrid inputs and supervised fine-tuning of the entire network. Under the linear evaluation protocol, our Spiking-ResNet18 achieves 59.4%, and scaling to Spiking-ResNet34 achieves 63.9%. Crucially, when evaluated under an equivalent fine-tuning protocol, our Spiking-ResNet18 reaches 70.21% and Spiking-ResNet34 reaches 72.31% (see Appendix C), surpassing (Singhal et al., 2024) by a substantial margin despite using only clean event inputs.

These results demonstrate that our dual-path SSL framework produces high-quality spiking representations that rival, and in some cases surpass, non-spiking self-supervised baselines.

## 4.2. Semi-Supervised Learning

Following (Zhai et al., 2019), we randomly sample 1% and 10% of the labeled training images from each dataset and fine-tune the entire pretrained model, without any additional regularization, on these small subsets. Table 2 reports accuracies under these extremely low-label regimes. With just 1% labels on ImageNet, Spiking-ResNet50 is comparable to Barlow-Twins by (52.6% vs. 55.0% Top-1). At 10% labels, it also almost matches Barlow-Twins (67.2% vs. 69.7% Top-1). On CIFAR-10, our Spiking ResNet-34 model exceeds Barlow-Twins by over 1.5% (75.1% vs. 73.6%) with 1% labels and by 2% (87.6% vs. 85.6%) with 10% labels. These results confirm superior performance of our spiking SSL framework in label-scarce scenarios.

## 4.3. Transfer Learning

**Image Classification**: We evaluate the transferability of our self-supervised features on three standard benchmarks: CIFAR-10, CIFAR-100, and Oxford Flowers-102. For each dataset, we conduct both i) linear evaluation, training a linear classifier on frozen features, and ii) full fine-tuning of the entire network. As shown in Table 3, Spiking-ResNet50 achieves linear evaluation scores of 90.6% on CIFAR-10, 70.3% on CIFAR-100, and 90.2% on Flowers-102, closely matching the Barlow Twins baseline (91.1%, 71.6%, and 92.1%, respectively). With end-to-end fine-tuning, Spiking-ResNet50 reaches 96.4% on CIFAR-10, 81.2% on CIFAR-100, and 95.5% on Flowers-102, which are also comparable to Barlow Twins (97.9%, 85.9%, 97.5%). Spikformer-8-512 shows similar accuracy across both evaluation settings, but

1indicates self-implementation due to the unavailability of reported results on these datasets.

*Table 3.* Transfer learning performance of the learned representations by our self-supervised learning pretrained on ImageNet-1K.

| | Method | CIFAR-10 | CIFAR-100 | Flowers |
|---|---|---|---|---|
| Linear evaluation | Barlow Twins-ResNet50[1] | **91.1** | **71.6** | **92.1** |
| | Ours-Spiking-ResNet50 | 90.6 | 70.3 | 90.2 |
| | Ours-Spikformer-8-512 | 89.9 | 70.1 | 90.8 |
| Fine-tune | Barlow Twins-ResNet50[1] | **97.9** | **85.9** | **97.5** |
| | Ours-Spiking-ResNet50 | 96.4 | 81.2 | 95.5 |
| | Ours-Spikformer-8-512 | 96.3 | 82.3 | 96.1 |

*Table 4.* Performance on COCO object detection and instance segmentation task with ImageNet-1K pretrained backbones.

| | COCO det | | | COCO instance seg | | |
|---|---|---|---|---|---|---|
| Method | $AP$ | $AP_{50}$ | $AP_{75}$ | $AP$ | $AP_{50}$ | $AP_{75}$ |
| Barlow Twins-ResNet50 | **39.2** | **59.0** | **42.5** | **34.3** | **56.0** | **36.5** |
| Ours-Spiking-ResNet50 | 37.8 | 57.4 | 41.1 | 33.3 | 55.4 | 35.5 |
| Ours-Spikformer-8-512 | 37.0 | 57.1 | 40.1 | 32.9 | 55.0 | 35.1 |

outperforms Spiking-ResNet50 at flowers task.

**Object Detection and Instance Segmentation**: We evaluate performance by fine-tuning our ImageNet pretrained backbones on COCO (Lin et al., 2014) using the Mask R-CNN framework (He et al., 2017). Following the setup in (Chen et al., 2020b), all models adopt the C4 backbone variant (Wu et al., 2019) and are trained using the standard $1\times$ schedule. As shown in Table 4, our Spiking-ResNet50 achieves an AP of 37.8 for bounding-box detection (vs. 39.2) and 33.3 for instance segmentation (vs. 34.3), slightly underperform the Barlow Twins ResNet50 baseline. The Spikformer-8-512 model also performs competitively, further demonstrating the strong transferability of our spiking self-supervised representations to dense prediction tasks.

## 4.4. Ablation Studies

**MixedLIF Training and Loss Functions**: To assess the effectiveness of our MixedLIF module and proposed temporal losses, we conduct an ablation study on CIFAR-10 using linear evaluation. Specifically, we compare: (i) the full MixedLIF model with dual-path activations against a baseline vanilla LIF model using a single spiking path with hard reset, and (ii) three self-supervised loss variants—Cross Temporal Loss (CTL), Boundary Temporal Loss (BTL), and Non-Cross Temporal Loss (NCTL), as summarized in Table 5. For Spiking-ResNet34 backbone, our MixedLIF with BTL achieves the highest accuracy of **85.6%**. Substituting BTL with CTL results in a minor decrease to 85.2%, while using NCTL leads to a larger drop to 82.5%. Disabling MixedLIF and training with vanilla LIF and BTL reduces accuracy to 83.0%; adding CTL improves it slightly to 84.1%, while combining vanilla LIF with NCTL yields 82.9%. Similar trends are observed for the transformer-based Spikformer backbone. MixedLIF with BTL achieves **84.3%**, with negligible degradation when using CTL (84.2%), and a more notable reduction with NCTL (82.3%). In contrast, vanilla LIF with BTL achieves 82.1%, while pairing it with CTL improves accuracy to 82.9%. The combina-

*Table 5.* Ablation study of neuron model and temporal loss by linear evaluation on CIFAR-10

| Backbone | Method description | Acc. (%) |
|---|---|---|
| Spiking-ResNet34 | MixedLIF + Boundary Temp Loss | **85.6** |
| | MixedLIF + Cross Temp Loss | 85.2 |
| | MixedLIF + Non-Cross Temp Loss | 82.5 |
| | LIF + Boundary Temp Loss | 83.0 |
| | LIF + Cross Temp Loss | 84.1 |
| | LIF + Non-Cross Temp Loss | 82.9 |
| Spikformer-4-384 | MixedLIF + Boundary Temp Loss | **84.3** |
| | MixedLIF + Cross Temp Loss | 84.2 |
| | MixedLIF + Non-Cross Temp Loss | 82.3 |
| | LIF + Boundary Temp Loss | 82.1 |
| | LIF + Cross Temp Loss | 82.9 |
| | LIF + Non-Cross Temp Loss | 81.7 |

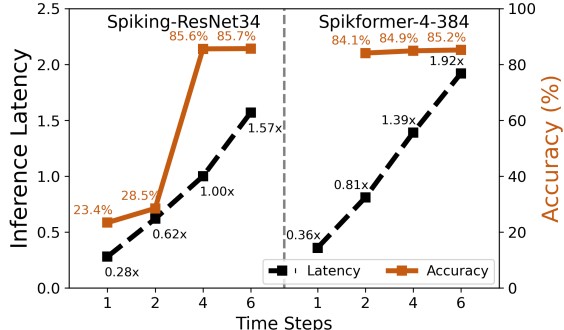

*Figure 3.* Top-1 accuracies (%) and inference latencies for different time steps under linear evaluation on CIFAR-10.

tion of vanilla LIF and NCTL produces the lowest result at 81.7%. These results collectively demonstrate that the SG-antiderivative neuron in MixedLIF is essential for stable and effective spiking self-supervised training, and that the Boundary Temporal Loss provides a favorable trade-off between computational efficiency and representation quality.

**Inference Latency**: Figure 3 shows how varying the number of time steps $T$ affects linear evaluation accuracy and inference latency for Spiking-ResNet34 and Spikformer-4-384, with $T=4$ on Spiking-ResNet34 as the latency baseline ($1\times$). At $T=1$, both models perform poorly but run with low latency ($0.28\times/0.36\times$). Increasing to $T=2$ improves accuracy ($28.5\%/84.1\%$) at moderate cost ($0.62\times/0.81\times$). Accuracy saturates at $T=4$ ($85.6\%/84.9\%$), while $T=6$ yields minimal gains with substantially higher latency ($1.57\times/1.92\times$). Thus, $T=4$ achieves the best balance. Details of energy-efficiency and neuromorphic deployment are provided in Appendix D and C respectively.

### 4.5. Training Efficiency

We evaluate the training latency of different loss functions on the CIFAR-10 dataset using two backbone architectures: Spiking-ResNet34 and Spikformer-4-384, as shown in Figure 4. For comparison, we introduce Non-Cross Temporal Loss (NCTL), which computes cross-correlations only between matching time steps across the two augmented views (i.e., $t$ of A with $t$ of B). All latency values are reported relative to NCTL on Spiking-ResNet34, which serves as the

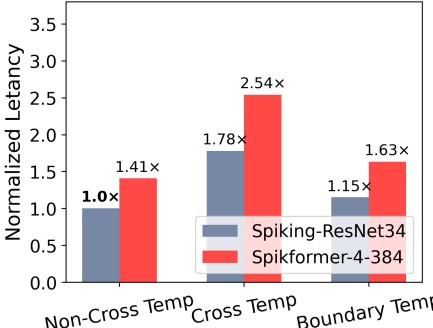

*Figure 4.* Training time comparison with Spiking-ResNet34 on CIFAR-10 between different loss functions. All values are normalized to the non-cross temporal loss.

baseline ($1\times$). For Spiking-ResNet34, the Cross Temporal Loss (CTL) incurs a significantly higher computational cost, increasing training time to $1.78\times$. In contrast, the Boundary Temporal Loss (BTL) results in only a slight increase ($1.15\times$), indicating minimal overhead. With the Spikformer-4-384 backbone, NCTL alone requires $1.41\times$ more time than the Spiking-ResNet34 baseline. CTL raises this to $2.54\times$, whereas BTL reduces the overhead to $1.63\times$. This highlights the training efficiency of BTL, offering a favorable trade-off between speed and performance, as shown below.

### 4.6. Comparison with Supervised SNN Baselines

To contextualize the effectiveness of our contrastive self-supervised framework, we compare our SSL-pretrained SNNs against fully supervised SNNs trained under identical architectural configurations (iso-architecture) and identical temporal dynamics ($t = 4$ timesteps). Table 6 reports results for both CIFAR-10 (Spiking-VGG16 and Spikformer-4-384) and ImageNet-1K (Spiking-ResNet50 and Spikformer-8-512). On ImageNet, our SSL-trained SNNs achieve top-1 and top-5 accuracies within a small margin of their supervised counterparts; notably, SSL even *outperforms* supervised training by 1.6% for Spiking-ResNet50, while trailing by 3.7% for Spikformer-8-512. To further evaluate Spikformer's performance, we perform fine-tuning with 300 epochs instead of linear evaluation, achieving 78.7% accuracy, which surpasses the supervised learning accuracy by 4.9% at iso-architecture. These results demonstrate that label-free pretraining can learn high-quality, transferable representations at scale.

On smaller datasets such as CIFAR-10, SSL-trained SNNs initialized from scratch naturally underperform supervised baselines, a well-known trend mirrored in the ANN literature, where SSL methods typically require large-scale data to learn semantically rich features. However, when transferring our ImageNet-pretrained SSL SNNs to CIFAR-10, we observe substantial gains: the resulting fine-tuned models

*Table 6.* Comparison of supervised and SSL-trained SNNs under iso-architecture settings using $T = 4$ timesteps. CIFAR-10 reports Top-1 accuracy; ImageNet-1K reports both Top-1 and Top-5 accuracy. FT indicates that the model's performance is obtained via full fine-tuning rather than linear evaluation.

| Dataset | Architecture | Training Type | Top-1 (%) | Top-5 (%) |
|---------|--------------|---------------|-----------|-----------|
| ImageNet-1K | Spiking-ResNet50 | Supervised | 67.9 | 88.6 |
| | Spiking-ResNet50 | SSL | 69.5 | 89.3 |
| | Spikformer-8-512 | Supervised | 73.8 | 93.3 |
| | Spikformer-8-512 | SSL | 70.1 | 89.9 |
| | Spikformer-8-512 (FT) | SSL | 78.7 | 94.1 |
| CIFAR-10 | Spiking-ResNet34 | Supervised | 92.9 | – |
| | Spiking-ResNet34 | SSL | 85.6 | – |
| | Spikformer-4-384 | Supervised | 95.2 | – |
| | Spikformer-4-384 | SSL | 84.9 | – |

exceed the performance of most existing supervised-trained SNNs (see Table 3). This further underscores the value of SSL as a scalable, label-efficient pretraining strategy. Even when downstream datasets are small, the representations learned through large-scale, unlabeled SSL pretraining provide strong generalization benefits that supervised-from-scratch SNNs cannot match.

## 5. Conclusions

We present a fully self-supervised learning framework for Spiking Neural Networks that leverages MixedLIF neurons and temporal alignment to learn rich, temporally structured representations. Our method incorporates a dual-path architecture and two novel training objectives, i) Cross Temporal Loss, and ii) Boundary Temporal Loss, that are designed to exploit the sequential dynamics of spike-based computation. Through extensive experiments across standard vision and neuromorphic benchmarks, we demonstrated that our models not only match but in some cases outperform non-spiking SSL baselines such as Barlow Twins. This stands in contrast to prior SNN work, which typically narrows but does not close the performance gap with ANNs. We attribute this performance gain to the rich temporal signals captured by our architecture during training, which help optimize SNNs more effectively than frame-based methods. We believe our work opens a promising path toward scalable and energy-efficient self-supervised neuromorphic learning systems. Further advances in scalable training methods and hardware integration will be critical to enabling widespread deployment of SSL SNNs in real-world applications.

## Impact Statement

This paper presents SpikeCLR, a fully self-supervised learning framework for Spiking Neural Networks (SNNs) aimed at energy-efficient visual representation learning. Our work contributes to neuromorphic computing and machine learning in ways that warrant broader societal consideration.

**Positive Societal Impacts.** SNNs deployed on neuromor-

phic hardware can substantially reduce inference energy consumption. Our estimates show more than $136\times$ lower compute energy compared to equivalent ANN-based self-supervised learning models. This has important implications for sustainable AI, enabling high-quality representation learning at the edge without reliance on energy-intensive server infrastructure. Applications such as robotics, autonomous sensing, and event-driven perception (e.g., gesture recognition and sign language interpretation) may benefit from label-efficient, low-power models that do not require large annotated datasets. In settings where labeled data is scarce or expensive to obtain, such as medical imaging, rare event detection, or neuromorphic sensor deployments, our approach may provide a practical pathway toward scalable representation learning.

**Limitations and Risks.** Although our framework reduces inference energy consumption, the pretraining stage remains computationally intensive, requiring substantial GPU resources. This training cost should be considered alongside long-term deployment benefits, particularly at scale. In addition, like other self-supervised vision models, Spike-CLR may inherit biases present in large-scale pretraining datasets such as ImageNet-1K, which could propagate to downstream applications if not carefully evaluated.

**Dual-Use Considerations.** Improvements in energy-efficient computer vision may also be applied to surveillance systems, autonomous decision-making, and other high-impact edge AI applications. While the primary goal of this work is scientific advancement and computational efficiency, such capabilities could be deployed in contexts that raise ethical concerns, including mass surveillance or autonomous weapons systems. Responsible downstream deployment and careful consideration of application contexts remain important.

Overall, this work highlights both the opportunities and responsibilities associated with scalable, energy-efficient, and label-free neuromorphic learning.

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

## A. Boundary Analysis

The *Boundary Temporal Loss* is motivated by the inherent temporal smoothness of spiking neural networks (SNNs), which arises from the leaky integration dynamics of neurons. For a standard Leaky Integrate-and-Fire (LIF) neuron, that does not spike with reset potential $V_{\text{reset}} = 0$, the membrane potential $H[t]$ evolves as:

$$H_t = \tau H_{t-1} + W X_t, \tag{10}$$

$$H_t = \tau^{T-1} H_1 + \sum_{k=0}^{T-1} \tau^{T-k} W X_k \tag{11}$$

This formulation shows that $H_t$ behaves as a low-pass filtered version of historical input, evolving smoothly across time. As a result, the intermediate activations $H_2, H_3, \ldots, H_{T-1}$ can be approximated as interpolations between the boundary states $H_1$ and $H_T$, with higher-order residuals. We further verified that adding intermediate timestep alignment provides marginal accuracy ($< 0.3\%$) but increases computation and memory access significantly, confirming the sufficiency of boundary sampling. Enforcing representation consistency at the start and end of the sequence thus implicitly regularizes intermediate steps due to the underlying dynamics. Moreover, these boundary states serve as critical temporal anchors. The initial state ($t = 1$) captures transient responses and high-frequency temporal features from the input, while the final state ($t = T$) encodes long-term dependencies through cumulative leaky integration. Together, they summarize the short- and long-term characteristics of spiking activity. Although this strategy does not explicitly supervise all time steps, its efficacy is supported by the sparse firing nature of SNNs. In our experiments, we measure an average spiking activity of approximately 23% across the network (see Figure. 5), indicating that neurons operate in subthreshold regimes most of the time. This ensures that hard resets, which could break temporal smoothness, occur infrequently. Consequently, the boundary representations retain most of the intermediate temporal information, enabling a biologically plausible and computationally efficient learning objective.

## B. Experimental Setup

**Data Preprocessing & Augmentations.** In our experiments, we pretrain on ImageNet, CIFAR-10 and three neu-

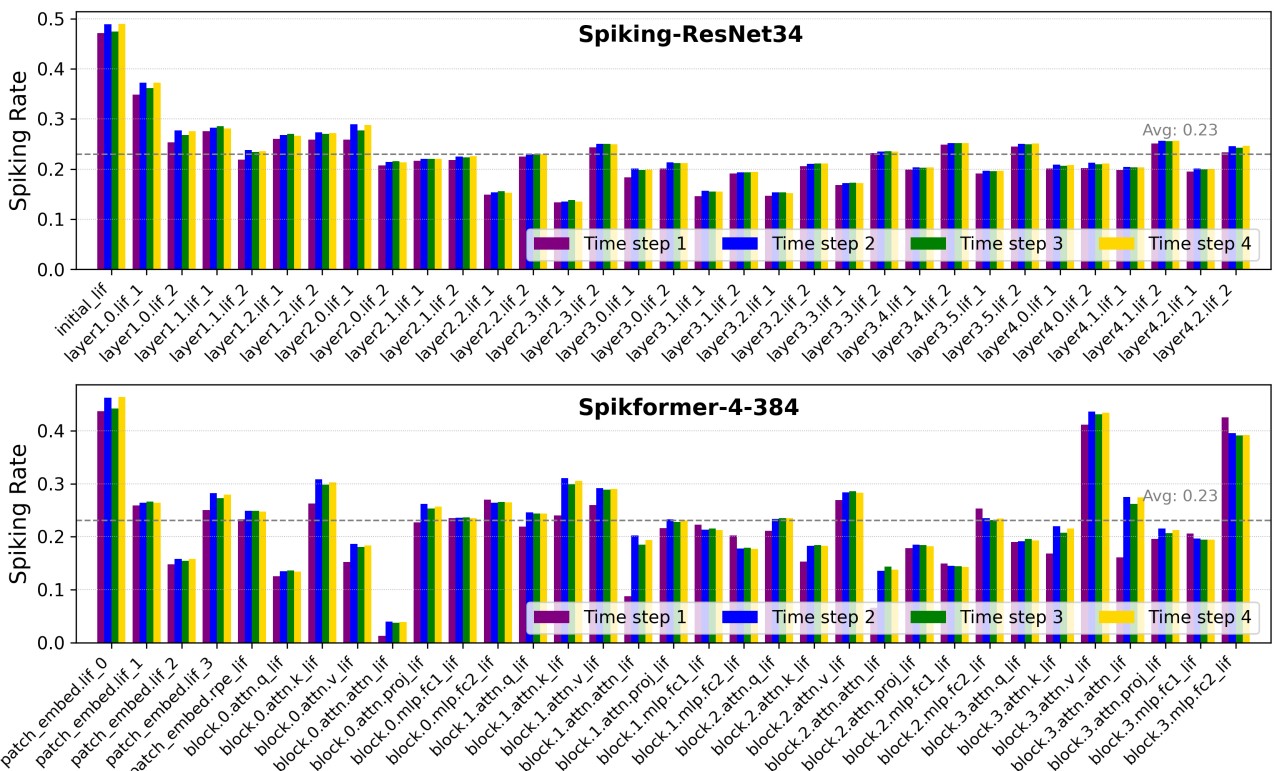

*Figure 5.* Layer-wise spiking rates over 4 time steps for Spiking-ResNet34 (top) and Spikformer-4-384 (bottom). Each cluster of four bars corresponds to one layer, with bar colors indicating time steps 1–4. The dashed horizontal line marks the average spiking rate across all layers and time steps.

romorphic datasets. For ImageNet and CIFAR-10 we apply the same spatial augmentations used in SimCLR (Chen et al., 2020b), BYOL (Grill et al., 2020a) and Barlow Twins (Zbontar et al., 2021): random crop and resize with horizontal flip, color distortion and Gaussian blur. Static images are then replicated into T time steps and resized to 224×224 for ImageNet, 32×32 for CIFAR-10.

For neuromorphic datasets, we replace color-based augmentations with a suite of event-specific transforms applied to the event frame tensor. Spatial augmentations include random horizontal and vertical flips, random rotation, and random crop with zero-padding. Temporal augmentations include time reversal with simultaneous polarity-channel swap and frame dropout. We further apply per-channel multiplicative polarity jitter, additive Gaussian background noise, and Gaussian blur applied uniformly across all timesteps. Finally, we resize the input to 128×128 for all neuromorphic datasets.

**Default Pretraining Settings.** We use Spiking-ResNet50 and Spikformer-8-512 (8 blocks, embedding dimension of 512) on ImageNet, each followed by a three-layer MLP projection head mapping to an 8192-dimensional space. For CIFAR-10, we adopt a modified Spiking-ResNet34 in which the original 7×7 stride-2 convolution is replaced

by a 3×3 stride-1 convolution with the subsequent max-pooling layer removed, and Spikformer-4-384 (4 blocks, embedding dimension of 384); both are paired with a lighter two-layer projection head (2048-dim hidden, 1024-dim output) for computational efficiency. For neuromorphic datasets, we employ Spiking-ResNet18, Spiking-ResNet34, and Spikformer-4-384, all with the same two-layer projection head as used for CIFAR-10.

All models are pretrained for 1000 epochs using momentum SGD (Ruder, 2016). On ImageNet-1K, the learning rate (LR) is selected from $\{1e-5, 1e-4, 1e-3\}$ via sweep with weight decay (WD) of $1.5e-6$. On CIFAR-10 and neuromorphic datasets, we sweep LR from $\{1e-4, 1e-3, 1e-2\}$ and WD of $1e-6$. To prevent early-stage gradient explosion, we apply a linear warmup over the first 10 epochs and then decay the learning rate according to a cosine schedule.

**Linear Evaluation.** Linear evaluation follows the protocol of (Zhang et al., 2016; Oord et al., 2018; Bachman et al., 2019). We freeze the pretrained backbone and train a linear classifier for 100 epochs using AdamW with a weight decay of $1e-6$ and a cosine annealing learning rate schedule. The base learning rate is set from $\{1e-5, 1e-4, 1e-3\}$ via sweep with a batch size of 256. During evaluation, inputs are augmented with random crop, resize, and optional

horizontal flip.

**Semi-Supervised Learning.** We train for 20 epochs using SGD with momentum (no weight decay). The base learning rate for backbones is set from $\{1e-5, 1e-4, 1e-3\}$ via sweep, and $5e-2$ for the final linear layer. We decay both rates by cosine annealing schedule. A batch size of 256 is used throughout.

**Transfer Learning.** For linear classifier, we extract frozen features from the ImageNet pretrained network and train a linear classifier with SGD with learning rate of $1e-3$ and weight decay of $1e-6$. No data augmentation is applied; only resizing is performed. For fine-tuning, we initialize the full network with pretrained weights on ImageNet and fine-tune for 100 epochs with batch size 256, using SGD with momentum $= 0.9$. During fine-tuning, only random resized crops and horizontal flips are applied as data augmentation. At test time, images are resized and center-cropped.

**Object Detection.** The experiments are built on the Detectron2 framework (Wu et al., 2019). We initialize Mask R-CNN (He et al., 2017) with our ImageNet pretrained Spiking-ResNet50 and Spikformer-8-512 backbones. We train the C4 variant on the COCO 2017 training split and evaluate on the validation split. All hyperparameters follow Detectron2's standard $1\times$ schedule, except that we set the base learning rate to 0.03.

## C. Extended Evaluation on Neuromorphic Datasets

**Controlled Comparison on CIFAR10-DVS.** Table 7 presents a controlled comparison with Singhal et al. (Singhal et al., 2024) under identical architecture and evaluation protocol ($T=10$, MixedLIF). Under the same Spiking-ResNet18 backbone and fine-tuning protocol, CTL achieves 70.21%, outperforming Singhal et al.'s 64.1% despite their use of hybrid event-frame inputs. Scaling to Spiking-ResNet34 with CTL further improves accuracy to 72.31%.

*Table 7.* Controlled comparison on CIFAR10-DVS ($T=10$, MixedLIF). Linear: frozen-encoder linear probing. Fine-tune: full network fine-tuning.

| Method | Architecture | Eval | Top-1 Acc. (%) |
|---|---|---|---|
| Singhal et al. (Singhal et al., 2024) | Spiking-ResNet18 | Fine-tune | 64.10 |
| BTL | Spiking-ResNet18 | Linear | 59.21 |
| BTL | Spiking-ResNet18 | Fine-tune | 68.34 |
| CTL | Spiking-ResNet18 | Linear | 59.35 |
| CTL | Spiking-ResNet18 | Fine-tune | 70.21 |
| BTL | Spiking-ResNet34 | Linear | 63.90 |
| BTL | Spiking-ResNet34 | Fine-tune | 68.97 |
| CTL | Spiking-ResNet34 | Linear | 64.21 |
| CTL | Spiking-ResNet34 | Fine-tune | **72.31** |

**Evaluation on Real-World Neuromorphic Datasets.** To further validate SpikeCLR on event-driven data, we conduct experiments on two natively captured event-camera

*Table 8.* Linear probing results on natively captured event-camera datasets (Spiking-ResNet18).

| Neuron Model | Loss | Top-1 Acc. (%) | |
|---|---|---|---|
| | | DVS128-Gesture ($T=16$) | ASL-DVS ($T=10$) |
| LIF | BTL | 76.21 | 96.21 |
| LIF | CTL | 77.34 | 97.37 |
| MixedLIF | BTL | 80.07 | 99.52 |
| MixedLIF | CTL | **80.82** | **99.93** |

datasets: DVS128-Gesture and ASL-DVS, in addition to the simulated CIFAR10-DVS benchmark that is converted from static images used in the main paper. Unlike CIFAR10-DVS, which is derived from static images and may not fully reflect real-world event-camera dynamics, both DVS128-Gesture and ASL-DVS are recorded directly with event sensors, providing a more realistic evaluation setting.

*DVS128-Gesture:* We evaluate on DVS128-Gesture (Amir et al., 2017) using Spiking-ResNet18 with $T=16$ and training settings identical to Appendix B.

*ASL-DVS:* ASL-DVS (Bi et al., 2019) is recorded using a DAVIS240c event camera capturing real human hand gestures for American Sign Language, providing a challenging and diverse benchmark for spatiotemporal representation learning.

Across all three event-based benchmarks (Table 8), CTL consistently outperforms BTL, in contrast to the static CIFAR-10 setting where the two perform comparably. This pattern is consistent with the CKA analysis in Appendix K: on event-driven data, each timestep receives distinct sparse input, producing non-monotonic representational trajectories that violate the boundary-interpolation assumption underlying BTL. CTL's exhaustive pairwise temporal alignment better captures this richer temporal structure, and the advantage becomes more pronounced at longer timestep horizons (e.g., $T=16$ on DVS128-Gesture). Together, these results demonstrate that SpikeCLR learns robust spatiotemporal representations that generalize across both simulated and natively captured event-camera data.

## D. Neuromorphic Deployment

Our spiking models are designed to be directly deployable on neuromorphic hardware such as Intel's Loihi 2. During inference, the surrogate-based continuous path (Path B) used for training is discarded entirely. Only the event-driven spiking path (Path A), based on LIF neurons, is retained, making our approach compatible with neuromorphic execution and low-power deployment.

We implemented both the Spiking ResNet34 and Spikformer architectures in Lava-DL (Team, 2023), Intel's software framework for deploying SNNs on Loihi, and evaluated them on CIFAR-10, and CIFAR10-DVS. As shown in Table 9, the accuracy degradation from PyTorch simulation to

*Table 9.* Accuracy (%) comparison between PyTorch simulation and Lava-DL deployment on Loihi.

| Model | Dataset | PyTorch Sim. | Lava-DL (Loihi) |
|---|---|---|---|
| Spiking-ResNet34 | CIFAR-10 | 85.6 | 84.9 |
| Spiking-ResNet34 | CIFAR10-DVS | 63.9 | 63.1 |
| Spikformer-4-384 | CIFAR-10 | 84.9 | 84.2 |
| Spikformer-4-384 | CIFAR10-DVS | 61.2 | 60.0 |

Lava-DL execution is minimal (typically 0.4–1.2%), demonstrating the robustness of our models under deployment constraints. To ensure compatibility with Loihi's fixed-point hardware, we fold batch normalization layers into the preceding convolutional or linear layers during Lava-DL implementation. This standard practice removes the need for separate batch norm execution during inference while preserving its representational effect. Despite minor quantization and precision-related constraints, the performance remains competitive, highlighting the deployability of our self-supervised SNNs in real-world edge settings.

## E. Training Efficiency

To quantify the computational overhead introduced by the additional forward path in MixedLIF, we benchmark per-batch training cost under identical settings (Tiny-ImageNet, $224 \times 224$ input resolution, batch size = 128) using the same Barlow Twins framework. We report training time, energy consumption, and peak GPU memory for ResNet34, Spiking-ResNet34-LIF, and Spiking-ResNet34-MixedLIF. For fairness, both spiking variants are evaluated with a single timestep ($t{=}1$).

*Table 10.* Per-batch training cost comparison across network variants. TT denotes training time and PM denotes peak memory.

| Method | TT (s) | Energy (J) | PM (MB) |
|---|---|---|---|
| ResNet34 | **0.878** | **145.699** | **6219.26** |
| Spiking-ResNet34-LIF ($t{=}1$) | 1.528 | 297.695 | 13702.16 |
| Spiking-ResNet34-MixedLIF ($t{=}1$) | **1.465** | **281.468** | **10750.12** |

Relative to the ANN baseline, Spiking-ResNet34-MixedLIF requires approximately $1.67\times$ training time and $1.93\times$ energy per batch. However, it remains consistently more efficient than the purely LIF-based model, reducing training time by 4.1%, energy consumption by 5.4%, and peak memory usage by 21.6%. These findings indicate that the two-path optimization strategy does not double the training cost; instead, the hybrid formulation leverages ANN pathways to reduce spike-driven computation and memory allocation, resulting in a substantially more favorable training profile compared to a full LIF architecture.

## F. Inference Efficiency

A key advantage of our proposed self-supervised SNN framework is its significantly lower energy consumption during inference, compared to dense ANN-based methods. While ANNs rely on multiply-and-accumulate (MAC) operations, SNNs compute using accumulate-only (AC) operations that are triggered by discrete spike events. This event-driven paradigm enables substantial reductions in energy consumption, particularly in sparsity-aware neuromorphic hardware, where inactive neurons and synapses incur no computational cost.

MAC operations in 32-bit fixed-point arithmetic consume approximately 3.1 pJ per operation in a 45nm CMOS process (Horowitz, 2014), whereas spike-driven accumulations typically require only 0.1 pJ, making them $31\times$ more efficient. For example, as shown in Table 11, a ResNet-34 model trained using Barlow Twins on CIFAR-10 involves roughly 3.6 GFLOPs per inference, yielding an estimated compute energy of 11.16 mJ. In comparison, our Spiking-ResNet34 processes inputs over 4 time steps with an average 23% spike activity, leading to approximately 828 million active accumulations. This translates to an estimated compute energy of just 0.082 mJ, more than $136\times$ lower than its ANN counterpart. Comparing our model to a purely LIF-based dual-path spiking baseline for SSL, we observe a 26% reduction (23% vs 29%) in spike activity on Spiking-ResNet34 and an 34% reduction (23% vs 31%) on Spikformer-4-384.

Beyond computation, memory access is a major contributor to energy consumption. ANNs must repeatedly load full activation maps and weights from memory, where each 32-bit SRAM access costs approximately 5 pJ, and DRAM access can exceed 100 pJ. In contrast, SNNs benefit from both activation sparsity and weight reuse. Since the same weights are applied across multiple time steps, they can be cached in local memory on neuromorphic or accelerator hardware, reducing redundant memory fetches. Additionally, spiking activations are sparse, so only a subset of neuron outputs are read or propagated at each step, further lowering bandwidth and memory energy usage on sparsity-aware systems. Moreover, since accumulation often occurs locally in neuromorphic implementations, write-back costs are reduced. As noted in Appendix C, during Lava-DL deployment, we fold batch normalization layers into the preceding convolution or linear layers, eliminating the need for runtime normalization without impacting performance.

Together, sparse compute, event-driven activations, and weight reuse, enable highly efficient inference. Our spiking models offer a compelling energy-performance trade-off, making them well-suited for real-time applications in edge and neuromorphic systems.

*Table 11.* Estimated inference-time compute for Barlow Twins and our spiking models on CIFAR-10. Energy is computed using 3.1 pJ/MAC for ANNs and 0.1 pJ/AC for SNNs based on (Horowitz, 2014). Spiking activity denotes the average proportion of active neurons over all the time steps.

| Model | Type | Time Steps | Spiking Activity | Active Ops (M) | Energy (mJ) |
|---|---|---|---|---|---|
| Barlow Twins (ResNet34) | ANN-SSL | 1 | 100% | 3600 | 11.16 |
| Spiking-ResNet34 | MixedLIF SNN-SSL | 4 | 23% | 828 | 0.082 |
| Spiking-ResNet34 | LIF SNN-SSL | 4 | 29% | 1044 | 0.103 |
| Spikformer-4-384 | MixedLIF SNN | 4 | 23% | 1569 | 0.157 |
| Spikformer-4-384 | LIF SNN | 4 | 31% | 2115 | 0.212 |

*Table 12.* Absolute Peak Memory (MB) on CIFAR-10 (Batch size = 256) for Spiking-ResNet34 and Spikformer-4-384.

| Method | Spiking-ResNet34 | Spikformer-4-384 |
|---|---|---|
| MixedLIF + NCTL | 22,582.19 | 30,309.64 |
| MixedLIF + BTL | 23,044.84 | 32,310.34 |
| MixedLIF + CTL | 25,039.39 | 36,403.98 |
| LIF + NCTL | 22,854.77 | 30,738.41 |

## G. Peak Memory Analysis

Table 12 compares the peak memory consumption of various models, their timestep configurations, and loss types. Notably, both Spikformer and ResNet34 models exhibit increased peak memory usage when trained with BTL and CTL loss functions, with CTL generally consuming the most. However, this increase remains within an acceptable range compared to non-cross-temporal training, which is the baseline for spiking SSL. Also, our MixedLIF neuron does not incur any additional peak memory overhead compared to the standard LIF neuron (LIF is used in both paths A and B), as they have similar activation dimensions for each layer in the network. Also, note that spiking models inherently consume more GPU memory than ANN models due to the need for temporal buffering and storage of multi-time-step representations.

Our experiments were conducted on CIFAR-10 with image resolution $32 \times 32$, using a batch size of 256 per GPU and a projection head dimension of 1024. For spiking models, we use 4 time steps, which naturally leads to approximately $\sim 4\times$ higher peak memory usage for models trained with BTL or non-contrastive loss compared to the ANN baseline trained with Barlow Twins. Notably, our MixedLIF + BTL configuration exhibits peak memory usage comparable to the baseline LIF model with non-contrastive loss, demonstrating that each of our proposed components (MixedLIF neurons and the BTL loss) incur no more memory overhead than a typical SNN trained with Barlow Twins. The CTL-trained spiking models require additional memory due to the storage of all 4 4 cross-correlation terms during temporal contrastive loss computation.

## H. Gradient Aggregation in MixedLIF

As shown in Figure 2 and detailed in Sections 3.2 of the paper, there is a single self-supervised loss computed using outputs from both Path A (spiking) and Path B (surrogate). While the two paths share identical network weights, they differ in their forward computations. Specifically,

1. Path A uses a threshold function during the forward pass, thereby forcing a surrogate gradient backpropagation with a clipped rectangular function.

2. Path B uses the antiderivative of this surrogate function (a clipped ReLU) during the forward pass, which results in smooth gradient signals.

Because of these differing forward computations, the gradients that each path produces during backpropagation are inherently different. These gradients are computed separately by detaching the complementary path in each case. This allows us to isolate clean gradient signals from both views. However, since both paths are processed in the same training pass and share parameters, their gradients are explicitly summed before the weight update step, rather than being treated separately or weighted. There is no weighting scheme involved. This design ensures that path A preserves the discrete spiking behavior critical for inference, and path B stabilizes training by providing dense gradient flow. The synergistic combination improves optimization, especially in the context of temporal self-supervised learning. We outline our algorithm for the gradient aggregation of the MixedLIF neuron below.

## I. Representation Learning in our Dual-Path Design of MixedLIF

To investigate whether the two paths in the MixedLIF neuron (spiking Path A and continuous Path B) learn different internal representations despite sharing weights, we conducted an empirical analysis using two standard similarity metrics: cosine similarity and KL divergence. We forward the same input (with identical augmentations) through both Path A and Path B. To avoid mutual influence during gradient computation, we detach one of the paths and only allow backpropagation through the other. This setup ensures a clean and fair measurement of representational difference without interference from simultaneous weight updates. Let $g_A$ and $g_B$ denote the intermediate gradients tensors obtained from Path A and Path B, respectively. Cosine similar-

**Algorithm 2** Gradient Aggregation for MixedLIF

---

1: **for** each batch $(X_A, X_B)$ in data loader **do**
2:    $X_A, X_B \leftarrow$ augment(batch) ▷ Two augmentations of input
3:    **Forward pass for Path A (spiking)**
4:    $out_A \leftarrow$ model.forward_spiking($X_A$)
5:    **Forward pass for Path B (non-spiking, anti-derivative path)**
6:    $out_B \leftarrow$ model.forward_surrogate($X_B$)
7:    **Compute SSL loss with gradients only through Path A**
8:    $loss_A \leftarrow$ compute_ssl_loss($out_A, detach(out_B)$)
9:    Backpropagate $loss_A$ and store gradients as $grad_A$
10:   Reset gradients
11:   **Compute SSL loss with gradients only through Path B**
12:   $loss_B \leftarrow$ compute_ssl_loss($detach(out_A), out_B$)
13:   Backpropagate $loss_B$ and store gradients as $grad_B$
14:   **Aggregate gradients from both paths**
15:   **for** each parameter $p$ in model **do**
16:      $p.grad \leftarrow grad_A + grad_B$
17:   **end for**
18:   **Update model weights**
19:   $optimizer.step()$
20:   Reset gradients
21: **end for**

---

ity is computed after flattening the tensors over the spatial, channel, and batch dimensions as shown below:

$$\cos\_\text{sim}(g_A, g_B) = \frac{g_A^\top g_B}{\||g_A\||_2 \||g_B\||_2}$$

To measure distributional differences, we also computed the KL divergence between the normalized gradient histograms of $g_A$ and $g_B$. Let $\mathbf{p}, \mathbf{q} \in \mathbb{R}^{50}$ denote the 50-bin histograms of $g_A$ and $g_B$, normalized with a small $\varepsilon = 1e-8$ added for numerical stability:

$$\mathbf{p} = \frac{\text{hist}(g_A)}{\sum_{i=1}^{B} \text{hist}(g_A)_i} + \varepsilon, \quad \mathbf{q} = \frac{\text{hist}(g_B)}{\sum_{i=1}^{B} \text{hist}(g_B)_i} + \varepsilon$$

The KL divergence between $\mathbf{p}$ and $\mathbf{q}$ can be computed using Eq. 12. On average, cosine similarity is around 0.45 across intermediate layers, indicating moderate alignment in direction between Path A and Path B representations. Despite the moderate cosine similarity (0.45), this directional alignment suggests that both paths may optimize toward similar functional goals in parameter space. The difference in the learning space between path A and path B may not imply conflict but rather complementary diversity. Notably, the moderate KL divergence of 0.87 further confirms that Path B provides a meaningful correction signal—its continuous gradient helps guide Path A's spiking behavior, while staying semantically aligned. Path B's continuous gradients serve as a corrective signal, smoothing out the noisy or sparse updates from the spike-driven Path A, and helping guide shared weights more stably.

## J. MixedLIF vs. LIF

**Dual-Path Setup.** In our framework, we explore two configurations for the dual-path encoder setup:

1. Both paths using LIF neurons (i.e., both use surrogate gradients based on clipped rectangular functions)

2. Path A using spiking LIF and Path B using a surrogate activation (MixedLIF) based on the antiderivative (clipped ReLU).

Both variants require two forward passes per batch (once through Path A and once through Path B), but their backward behavior differs. In dual-LIF, both paths produce sparse activations, enabling sparse gradient computation during backprop. In MixedLIF, Path B uses a non-sparse, differentiable surrogate path, making both the forward and backward pass denser and slightly more compute-heavy. However, the additional overhead of the time incurred during the forward and backward passes in MixedLIF may be negligible as evidenced by our empirical results shown below, because modern GPUs may not leverage irregular spike sparsity very well.

Additionally, the choice of temporal loss directly impacts both compute cost and memory complexity:

1. Non-Cross Temporal Loss (NCTL) computes independent contrastive losses at each time step, resulting in O(T) scaling, and $T$ loss terms.

2. Boundary Temporal Loss (BTL) computes loss only between the first and last time steps, avoiding iteration over all time steps and keeping complexity constant with respect to T. It incurs $\binom{4}{2} = 6$ cross-correlation terms to calculate the loss value.

3. Cross Temporal Loss (CTL) computes pairwise contrastive loss across all $(t_1, t_2)$ pairs, leading to O(T$^2$) computational complexity and the largest training cost. It incurs $\binom{2T}{2} = T(2T-1)$ cross-correlation terms to calculate the loss value, which is 28 for T=4.

These differences are further amplified in the dual-path setup, where loss is computed separately for each path and gradients are summed.

Our theoretical complexity estimates align closely with empirical results. Notably, forward pass durations remain consistent across configurations, suggesting that the structural

*Table 13.* Theoretical Training Time Complexity

| Configuration | Forward | Backward | Loss | Explanation |
|---|---|---|---|---|
| Dual LIF + NCTL | 2× | ∼2× | ∼T× | Sparse gradients in both paths. Efficient training. |
| MixedLIF + NCTL | ∼2× | ∼2× | ∼T× | Path B has dense gradients from anti-derivative. |
| MixedLIF + BTL | 2× | ∼2× | ∼6× | Loss only at t=0 and t=T. Constant cost w.r.t T. |
| MixedLIF + CTL | 2× | ∼2× | ∼T(2T-1)× | Pairwise temporal loss over T×T steps. |

*Table 14.* Per-Batch Training Time on ImageNet (Batch Size = 128)

| Method | Forward | Backward | Loss | Total |
|---|---|---|---|---|
| Dual LIF + NCTL | 0.382s | 1.024s | 0.003s | 1.409s |
| MixedLIF + NCTL | 0.381s | 1.024s | 0.003s | 1.408s |
| MixedLIF + BTL | 0.378s | 1.089s | 0.165s | 1.632s |
| MixedLIF + CTL | 0.376s | 1.349s | 0.814s | 2.539s |

differences (e.g., MixedLIF vs. LIF) do not significantly impact the forward path latency. In contrast, loss computation times differ substantially, with the CTL incurring significantly higher overhead (∼4.9×) compared to boundary or non-cross temporal loss as shown above. For simpler datasets, such as CIFAR10, a similar trend can be observed as shown in Fig. 3 in the paper (CTL incurs 2.3-2.4x higher training time compared to BTL). This confirms that loss formulation, rather than neuron model choice, is the dominant factor in training efficiency. Therefore, practitioners seeking efficiency–accuracy trade-offs may prefer Boundary Temporal Loss as a middle ground, balancing temporal structure with low training complexity.

**Convergence.** To analyze the optimization behavior of the proposed MixedLIF design, we conduct a controlled comparison against a standard LIF-based spiking encoder. We train two Spiking-ResNet34 models on CIFAR-10 using the Barlow Twins objective, timestep $T = 4$, boundary temporal loss, and the default training configuration. The only difference between the two settings is the activation formulation: the baseline uses dual LIF paths, whereas MixedLIF replaces Path B with a clipped-ReLU surrogate antiderivative. All other architectural components, data augmentation, optimizers, and hyperparameters remain identical, ensuring a fair comparison.

As shown in Fig. 6, CIFAR-10's relatively small scale allows both models to fully converge within 500 epochs. During the early training stage ($< 40$ epochs), the dual-LIF model exhibits a slightly faster loss decay. However, after approximately 40 epochs, the differentiable surrogate pathway in MixedLIF begins to stabilize optimization by mitigating gradient mismatching, leading to a consistently lower training loss. At epoch 500, MixedLIF achieves a final loss of **70.62**, compared to **75.47** for the LIF variant, demonstrating improved convergence quality.

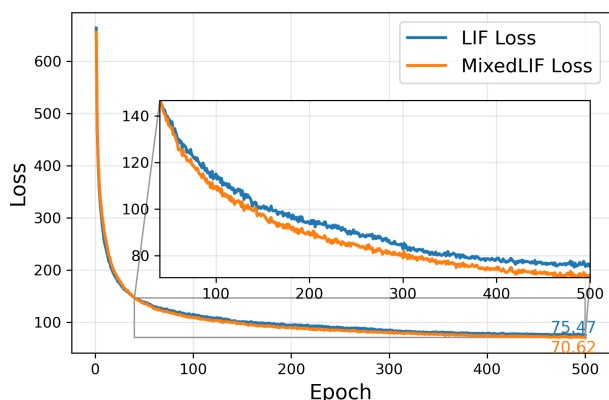

*Figure 6.* Training loss comparison between LIF and MixedLIF on CIFAR-10 using Spiking-ResNet34. Both models are trained with timestep $T = 4$ and identical Barlow Twins configurations. The inset zoom illustrates the point at which MixedLIF surpasses LIF in convergence rate and final loss.

# K. CTL vs. BTL: Per-Step Representational Analysis

We analyze per-step representational similarity to explain the divergent behavior of BTL and CTL across static and event-driven datasets.

**Static Datasets.** For static inputs such as CIFAR-10, the same image is replicated across all timesteps. LIF dynamics induce smooth temporal evolution in which intermediate membrane states closely approximate interpolations between the boundary timesteps. BTL leverages this structure by aligning only the first and last timesteps, thereby capturing nearly all temporal information at a fraction of CTL's computational cost, which explains why BTL performs comparably to, or marginally better than, CTL on CIFAR-10.

To quantify this, we compute per-timestep KL divergence between the latent features of a CTL-trained and a BTL-trained Spiking-ResNet34 over the CIFAR-10 test set. Specifically, for each timestep $t$, we extract feature vectors $z_t^{\text{CTL}}$ and $z_t^{\text{BTL}}$ and compare their distributions via histogram-based

KL divergence:

$$D_{\mathrm{KL}}(\mathbf{p} \parallel \mathbf{q}) = \sum_{i=1}^{50} p_i \log\left(\frac{p_i}{q_i}\right) \qquad (12)$$

where $\mathbf{p}$ and $\mathbf{q}$ are normalized activation histograms from each model at a given timestep. As shown in Table 15, the per-timestep KL divergences remain stable across all timesteps (range: 0.251–0.260), confirming that CTL and BTL produce globally similar representations on static data.

*Table 15.* Per-timestep KL divergence between CTL and BTL feature distributions on CIFAR-10 (Spiking-ResNet34, MixedLIF, $T=4$).

| Timestep | Mean KL | Std. Dev. |
|---|---|---|
| 1 | 0.252 | 0.104 |
| 2 | 0.260 | 0.108 |
| 3 | 0.257 | 0.104 |
| 4 | 0.259 | 0.107 |

**Event-Driven Datasets.** For event-driven inputs such as CIFAR10-DVS, each timestep receives a distinct sparse event frame, producing non-monotonic representational trajectories that violate the boundary-interpolation assumption underlying BTL. In fast or complex motion scenarios, boundary frame distributions diverge substantially, and intermediate timesteps encode information that cannot be recovered from the boundaries alone.

To quantify this effect, we compute CKA heatmaps for CIFAR10-DVS with Spiking-ResNet18 ($T=10$), shown in Figure 7. The results reveal markedly greater temporal diversity than observed on static data. Specifically, BTL's inter-timestep CKA drops to approximately 0.5 for intermediate pairs, whereas CTL maintains pairwise similarity above 0.87 throughout the full temporal sequence. A cross-model CKA of 0.76 further indicates that the two objectives yield representationally distinct solutions. CTL's exhaustive pairwise alignment better preserves this richer temporal structure, accounting for its superior performance on event-driven benchmarks where boundary-only supervision is insufficient.

## L. Comparison with Other LIF Variants

We compare our proposed **MixedLIF** neuron with other popular LIF-based neuron models: i) **LIF**: Standard Leaky Integrate-and-Fire neuron, ii) **PLIF** (Deng et al., 2022): Trainable leak, iii) **ALIF** (Bellec et al., 2018b): LIF neuron with an adaptive threshold that increases after each spike and decays over time, and iv) **IF**: Integrate-and-Fire neuron with no leak term ($\tau = 0$). These models are evaluated under the same dual-path SSL setup with BTL. Unlike MixedLIF, these alternative neurons use the

same neuron type in both paths. Thus, they cannot benefit from the full-precision surrogate gradient path provided by

*Table 16.* Comparison of MixedLIF with alternative neuron models on CIFAR-10 under the BTL loss. The backbone used is Spiking-ResNet34.

| Neuron Model | Linear Eval Top-1 (%) |
|---|---|
| MixedLIF | 85.6 |
| LIF | 83.0 |
| PLIF | 83.4 |
| ALIF | 82.9 |
| IF | 81.3 |

MixedLIF, which stabilizes training and provides temporally consistent gradient signals across paths. As a result, their accuracy drops significantly, with declines exceeding **2%** compared to MixedLIF, demonstrating that MixedLIF's dual-path design is essential for scaling SNNs to large-scale self-supervised learning tasks.

As shown in Table 16, replacing MixedLIF with any of these alternatives results in a noticeable accuracy drop, confirming that MixedLIF is critical for achieving high performance in SSL settings. This highlights the importance of our design, which leverages the temporal dynamics of the non-spiking path to improve representation learning while preserving spiking sparsity during inference.

## M. Effect of Reset Mechanisms on Temporal Consistency in SSL

A key factor affecting the stability of self-supervised learning (SSL) in SNNs is the neuron reset mechanism. Standard *soft-reset* LIF neurons retain a residual membrane potential after emitting a spike, which introduces variability in the post-spike state. While this behavior is often beneficial for supervised tasks, we observe that in SSL it leads to inconsistent temporal statistics across augmented views. In particular, soft reset produces higher variance in spike-count distributions, causing fluctuations in the temporal dynamics that downstream contrastive objectives must align.

In contrast, the *hard-reset* mechanism used in MixedLIF directly resets the membrane potential to a fixed value after each spike, removing residual state differences between augmentations. This enforces more consistent temporal evolution across views and leads to significantly more stable spike-count distributions. As shown in Table 17, this stabilization improves the consistency of the SSL loss and yields a **+1.15%** accuracy improvement on CIFAR-10. Thus, by enforcing consistent post-spike states and reducing augmentation-induced variability, the hard-reset formulation in MixedLIF provides a more stable temporal signal that better supports cross-view learning.

## N. Visualization

To further understand the impact of different training strategies on learned representations, we visualize the feature

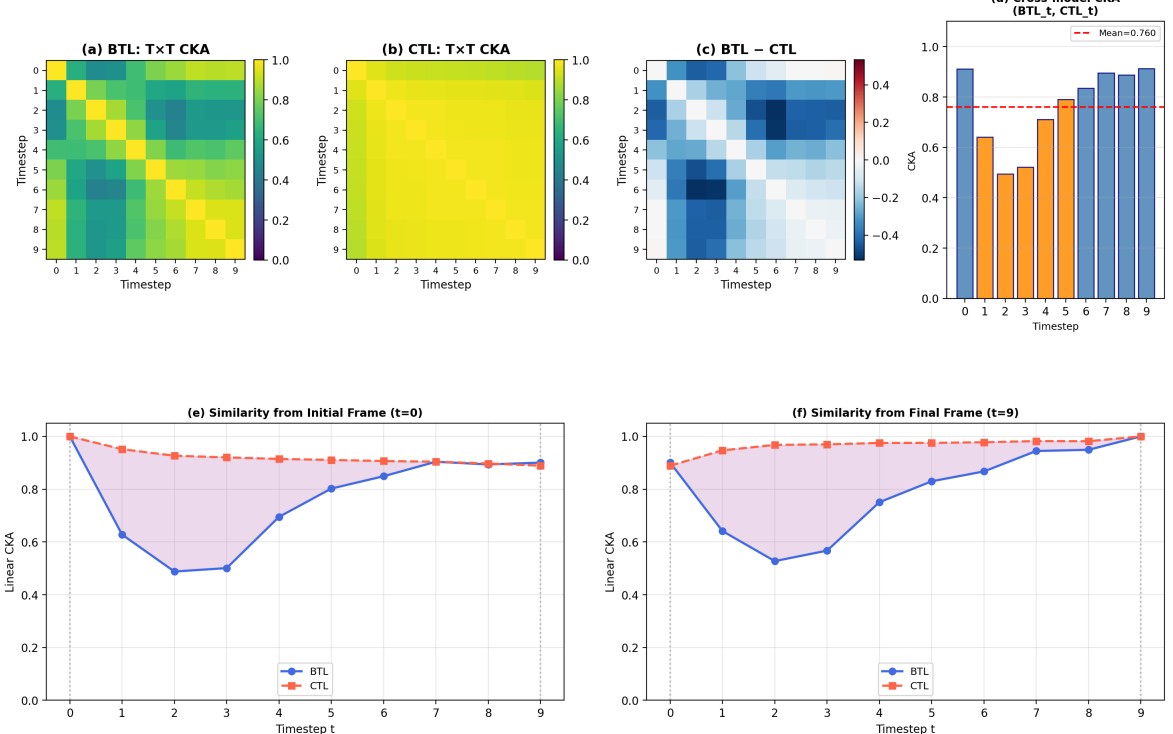

*Figure 7.* CKA heatmaps comparing BTL and CTL on CIFAR10-DVS (Spiking-ResNet18, $T$=10). CTL maintains high pairwise similarity ($> 0.87$) across all timestep pairs, whereas BTL degrades to $\sim$0.5 for intermediate steps, revealing the limitations of boundary-only alignment on event-driven data.

*Table 17.* Impact of reset mechanism on temporal consistency and SSL performance for Spiking-VGG16 and Spikformer-4-384. Spike-count variance is computed across augmentations and normalized to the soft-reset baseline.

| Model | Reset Type | Spike Count Var. | SSL Loss Var. | Acc. (%) |
|---|---|---|---|---|
| Spiking-VGG16 | Soft Reset | 1.00× | 1.00× | 81.1 |
| Spiking-VGG16 | Hard Reset | 0.70× | 0.78× | 81.9 |
| Spikformer-4-384 | Soft Reset | 1.00× | 1.00× | 83.4 |
| Spikformer-4-384 | Hard Reset | 0.51× | 0.63× | 84.9 |

distributions using t-SNE (Van der Maaten & Hinton, 2008) projections. Figure 8 presents the t-SNE visualizations of the representations learned by different configurations at each time step. The baseline model using MixedLIF and Boundary Temporal Loss (Figure 8a) shows well-separated clusters with clear boundaries between classes, indicating strong discriminative features across all time steps. This aligns with its superior classification performance (85.6%) on the CIFAR-10 dataset.

The model trained with vanilla LIF (Figure 8c, d, and e) exhibits less defined clustering, particularly in earlier time steps. This suggests that the MixedLIF activation function helps establish more discriminative features in the temporal

processing pipeline. The configuration using Non-Cross Temporal Loss model (Figure 8b and e) shows the least separation between clusters, consistent with its comparatively lower performance (82.9%). The visualization reveals that loss formulation contribute significantly to the quality of learned representations.

These visualizations collectively demonstrate how different combinations of activation functions and loss formulations influence the feature space organization across time steps, providing insights into why proposed MixedLIF and Boundary/Cross Temporal Loss achieve better classification performance than others.

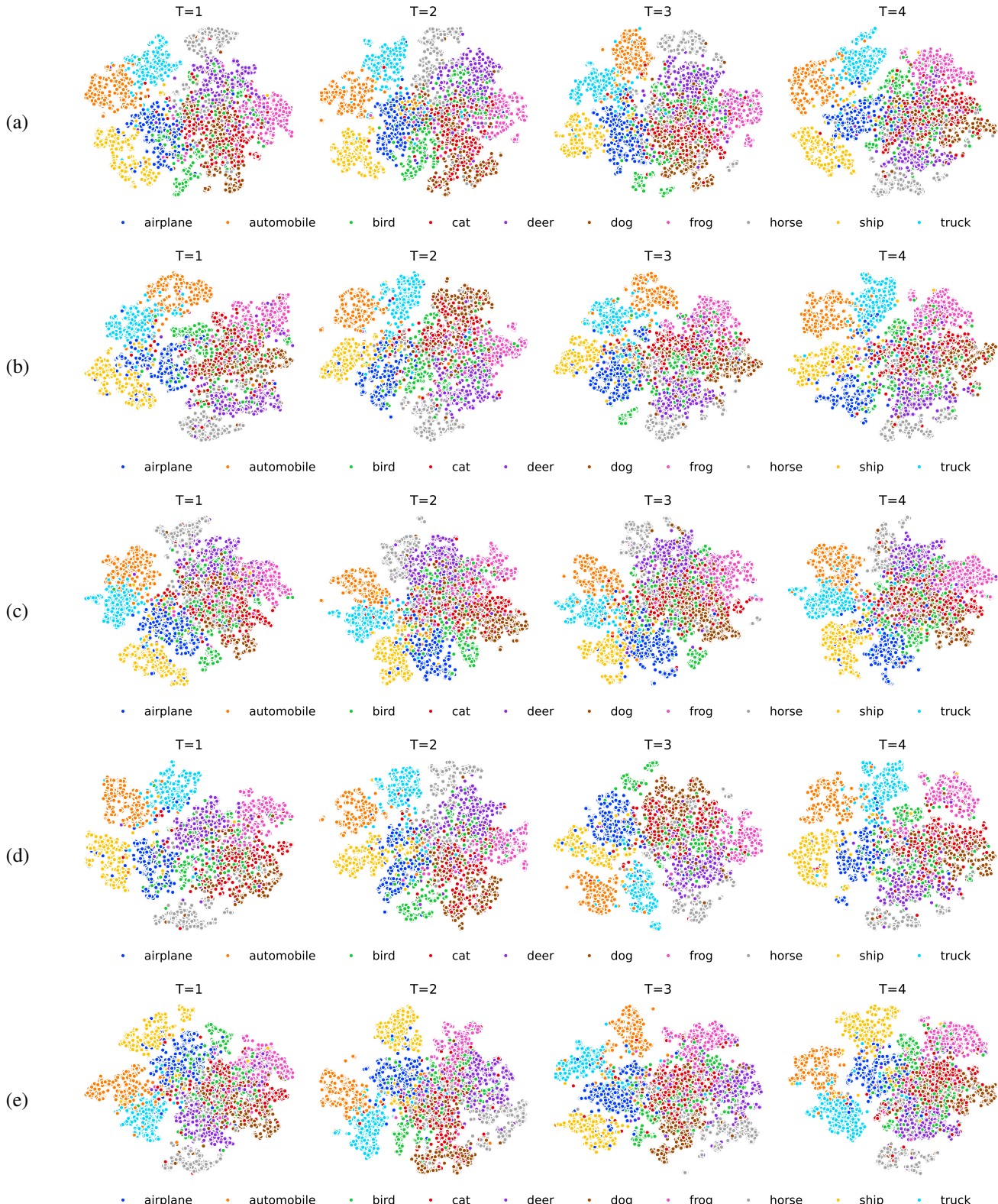

*Figure 8.* t-SNE visualization of Spiking-ResNet34 representations on CIFAR-10 for each ablation setting. (a) MixedLIF + Boundary Temporal Loss (baseline, 85.6% acc.), (b) MixedLIF + Non-Cross Temporal Loss (82.5% acc.), (c) LIF + Boundary Temporal Loss (83.0% acc.), (d) LIF + Cross Temporal Loss (84.1% acc.), and (e) LIF + Non-Cross Temporal Loss (82.9% acc.). Each row shows the progression of feature representation quality across time steps, with columns representing increasing time steps from left to right. Colors represent different CIFAR-10 classes. Note how higher-performing configurations display more distinct class clustering, especially in later time steps.

