# OpenReview forum: "SpikeCLR: Self-Supervised Contrastive Learning for Visual Representations with Spiking Neural Networks"
_ICML.cc/2026/Conference — ICML 2026 regular_

### Official Review · Reviewer_nQES · 2026-03-10

**Soundness:** 3
**Presentation:** 3
**Significance:** 3
**Originality:** 3
**Overall Recommendation:** 4
**Confidence:** 2

**Summary:**

This paper proposes SpikeCLR, a framework for training SNNs using self-supervised learning. The authors address the difficulty of training SNNs without labels by introducing a "MixedLIF" neuron, which uses two paths during training: a standard spiking path and a continuous surrogate path to help with gradient flow. They also introduce two loss functions, Cross Temporal Loss and Boundary Temporal Loss, designed to align the model’s outputs across different time steps. The method is tested on datasets like CIFAR-10 and ImageNet-1K using both ResNet and Transformer-based architectures.

**Compliance With Llm Reviewing Policy:**

Affirmed.

**Key Questions For Authors:**

**Question 1:** facilitate gradient flow. Does the reliance on Path B's gradients risk the model simply "inheriting" ANN-like features rather than learning intrinsically spiking-optimal representations?

**Question 2:** Have the authors observed any representation collapse or severe distributional shifts if soft resets were used instead of hard resets?

**Question 3:** The Boundary Temporal Loss (BTL) is highly efficient and performs well, sometimes slightly outperforming the Cross Temporal Loss. Could you add a brief intuition to the text regarding why the intermediate time steps contribute so little to the representational quality on static datasets compared to the boundaries?

**Limitations:**

I can't find the limitation of the worked mentioned in the manuscript.

**Strengths And Weaknesses:**

**Soundness:** The methodology appears consistent with standard practices in the field. The authors evaluate their approach using linear probes, semi-supervised fine-tuning, and transfer learning to tasks like object detection. The use of hard resets in the MixedLIF module is intended to keep the two training paths synchronized. While the results are documented, more specific details on the tuning of the lambda hyperparameter in the loss functions would be helpful for those trying to replicate the work.

**Presentation:** The paper is organized and easy to follow. The diagrams illustrate how the dual-path neuron works and how the temporal losses are calculated. The authors define the limitations of current SNN training and explain how their proposed modules are intended to function within a contrastive learning setup.

**Significance:** This work shows that it is possible to apply self-supervised learning to SNNs on a dataset as large as ImageNet-1K. While other SSL methods for SNNs exist, many have stayed limited to smaller or neuromorphic-only datasets. By reaching competitive accuracies on standard benchmarks, the paper demonstrates that SNNs can learn useful features without manual labels in a more scalable way than previously shown. However, the work could be evaluated on more datasets.

**Originality:** The main contribution is the MixedLIF neuron design, which attempts to solve the gradient mismatch in SNNs by using the antiderivative of a surrogate function as a continuous training path. Additionally, the introduction of temporal-specific losses (CTL and BTL) to specifically handle the time dimension in SNNs differentiates this from standard frame-based self-supervised methods.

---

> ### Author Rebuttal · Authors · 2026-03-31
>
> Dear reviewer nQES,
>
> Thank you for your review, which has helped improve our work. We address your comments (W) and questions (Q) below.
>
> **W1**: More details on tuning the $\lambda$ hyperparameter.
>
> We use $\lambda$=5e-3 across all settings, following the original Barlow Twins paper. We also conducted a hyperparameter sweep over $\lambda\in$ {1e-2, 5e-3, 1e-3, 5e-4}, and found that $\lambda$=5e-3 consistently yielded the best performance.
>
> **Q1**: "inheriting" ANN-like features with Path B's gradients?
>
> By design, our framework guards against this risk through equal-weight gradient aggregation. As specified in Algorithm 1 (line 14), the shared parameters are updated as θ <- θ − \eta(g_A + g_B), where g_A and g_B are the gradients from the spiking and continuous paths respectively, with no weighting in favor of either path. Path A's spiking gradients therefore exert equal influence on the weight update as Path B's continuous gradients. The resulting optimization trajectory is not dominated by ANN-like signals but is instead a balanced fusion of both discrete spiking dynamics and smooth surrogate guidance.
>
> Empirically, this balance produces better optimization than either path alone. As shown in Appendix J (Figure 6), MixedLIF achieves a final training loss of 70.62 compared to 75.47 for the dual-LIF variant, demonstrating that the combined gradient signal finds a better minimum. Crucially, the resulting representations remain spiking-optimal at inference: only Path A is retained, and the average spiking activity is approximately 23% (Table 9, Figure 5), which is actually sparser than the 29% observed in the dual-LIF baseline. If Path B were forcing ANN-like dense representations onto the shared weights, we would expect higher spiking rates, not lower. The lower spiking activity under MixedLIF indicates that Path B's gradient stabilization helps the spiking path learn more efficient, sparser representations rather than mimicking ANN behavior.
>
> **Q2**: Representation collapse or severe distributional shifts with soft reset?
>
> Yes, we observed measurable degradation with soft resets. As reported in Table 15 (Appendix N), switching from hard to soft reset consistently reduces accuracy across both architectures: from 81.9% to 81.1% on Spiking-VGG16 and from 84.9% to 83.4% on Spikformer-4-384. The Spikformer backbone is more severely affected (1.5% drop vs 0.8%), which we attribute to its deeper attention-based architecture amplifying distributional shifts across layers.
>
> Under soft reset, a residual membrane potential persists after each spike, and this residual differs between Path A (discrete spike output) and Path B (continuous activation). These residual differences accumulate layer by layer, causing the post-spike state distributions of the two paths to progressively diverge. This is reflected in the spike-count variance, which increases by 30% on Spiking-VGG16 and 49% on Spikformer-4-384 under soft reset relative to hard reset. The SSL loss variance increases correspondingly (22% and 37%), indicating that the contrastive objective receives inconsistent targets across augmented views. While we did not observe complete representation collapse (accuracies remain above 81%), the systematic degradation confirms that soft reset introduces sufficient distributional instability to impair self-supervised training. Hard reset eliminates this issue by enforcing a deterministic post-spike state ($V_{reset}$=0) in both paths, keeping their temporal dynamics synchronized and the cross-correlation targets well-aligned.
>
> This also connects to the theoretical basis (Appendix A), which relies on the temporal smoothness of LIF dynamics for boundary-only alignment to be sufficient. Hard reset preserves this smoothness, whereas soft reset's residual potentials disrupt the low-pass filtering property, undermining BTL's interpolation assumption.
>
> **Q3**: Effect of BTL on the representational quality of static datasets compared to the boundaries?
>
> First, the optimal number of timesteps for static datasets is small. As shown in Figure 3, accuracy saturates at T=4, with T=6 providing negligible gain. With only two intermediate states, CTL's exhaustive pairwise alignment offers minimal additional signal beyond what BTL already captures at the boundaries. Second, on static datasets the same image is replicated at every timestep, so LIF dynamics produce a smooth trajectory from an initial transient (t=1) toward a steady state (t=T). As shown in Appendix A (Eqs. 10-11), H_t converges exponentially toward a fixed point H*=WX/(1−$\tau$), making intermediate representations near-interpolations of the boundaries with minimal unique information. Boundary alignment thus implicitly regularizes intermediate states through the underlying dynamics. This is corroborated by low spiking activity (~23%, Figure 5), indicating neurons predominantly operate in subthreshold regimes where temporal smoothness is preserved.

---

> > ### Author Rebuttal · Reviewer_nQES · 2026-04-03
> >
> > I thank the authors for their detailed rebuttal and their efforts to address the concerns raised during the review process. After carefully reading the response, I will be maintaining my score.
> >
> > The core promise of contrastive SSL is the ability to learn robust, generalized spatio-temporal representations from complex, unlabeled data. Three of the four evaluated datasets (N-MNIST, N-Caltech101, and CIFAR10-DVS) are converted from static images and inherently lack the rich, real-world temporal dynamics that event cameras capture. While testing on four datasets might be standard for supervised SNN papers, an SSL framework requires broader and more complex empirical validation to prove it doesn't just overfit to the limited spatio-temporal properties of these specific, restricted benchmarks.
> >
> > To convincingly demonstrate the generalizability and data-efficiency claims of SpikeCLR, the framework would need to be evaluated on a larger, more diverse suite of native event datasets (or at a larger scale) to truly showcase its transfer learning capabilities. While this paper presents a highly interesting direction for SNNs, I believe the current experimental scope is too narrow to confidently validate the effectiveness of the proposed contrastive method.

---

> > > ### Author Response · Authors · 2026-04-08
> > >
> > > We thank the reviewer for the thoughtful feedback. We agree that native event datasets with genuine temporal dynamics are essential for validating our SSL framework. To directly address this concern, we have added evaluation on two natively captured event datasets that were **not** converted from static images:
> > >
> > > **DVS128 Gesture** [1] contains 1,342 recordings of 11 hand gesture classes performed by 29 subjects under 3 lighting conditions, captured natively by a DVS128 event camera. The temporal dynamics arise from real human motion, not artificial saccades applied to static images.
> > >
> > > **ASL-DVS** [2] contains approximately 100,800 samples across 24 American Sign Language letter classes, recorded using a DAVIS240c event camera capturing real hand gestures. This is one of the largest natively captured event classification benchmarks available.
> > >
> > > *DVS128 Gesture results (Spiking-ResNet18, T=16, linear probing):*
> > > | Method | Loss | Top-1 Acc. (%) |
> > > |---|---|---|
> > > | LIF | BTL | 76.21 |
> > > | LIF | CTL | 77.34 |
> > > | MixedLIF | BTL | 80.07 |
> > > | MixedLIF | CTL | 80.82 |
> > >
> > > *ASL-DVS results (Spiking-ResNet18, T=10, linear probing):*
> > > | Method | Loss | Top-1 Acc. (%) |
> > > |---|---|---|
> > > | LIF | BTL | 96.21 |
> > > | LIF | CTL | 97.37 |
> > > | MixedLIF | BTL | 99.52 |
> > > | MixedLIF | CTL | 99.93 |
> > >
> > > Two consistent findings emerge across both native event datasets. First, MixedLIF outperforms vanilla LIF by a significant margin (3.5–3.9% on DVS128 Gesture and 2.6-3.3% on ASL-DVS), confirming that our dual-path design is essential for stable self-supervised training on real event data. Second, CTL consistently outperforms BTL on these native event datasets, in contrast to static datasets where they perform comparably. This validates that CTL better captures the genuine temporal dynamics present in natively captured event streams, where intermediate timesteps carry unique information that boundary-only alignment misses.
> > >
> > > Together with our ImageNet-1K pretraining, transfer to COCO detection and segmentation, and downstream evaluation on CIFAR-10, CIFAR-100, and Flowers-102, we believe the framework is now validated across both large-scale static benchmarks and natively captured event data with real temporal dynamics. We will incorporate all these new results into the revised manuscript.
> > >
> > > **References**
> > >
> > > [1] Amir, A., Taba, B., Berg, D., Melano, T., McKinstry, J., Di Nolfo, C., Nayak, T., Andreopoulos, A., Garreau, G., Mendoza, M., et al. (2017). A Low Power, Fully Event-Based Gesture Recognition System. In Proceedings of the IEEE Conference on Computer Vision and Pattern Recognition (CVPR).
> > >
> > > [2] Yin Bi, Aaron Chadha, Alhabib Abbas, Eirina Bourtsoulatze and Yiannis Andreopoulos, 'Graph-based Object Classification for Neuromorphic Vision Sensing', IEEE Conference on Computer Vision (ICCV), Oct.17 - Nov,2, 2019, Seoul, Korea

---

### Official Review · Reviewer_Y1Kf · 2026-03-11

**Soundness:** 3
**Presentation:** 3
**Significance:** 3
**Originality:** 3
**Overall Recommendation:** 4
**Confidence:** 4

**Summary:**

This paper introduces SpikeCLR, the first fully self-supervised learning (SSL) framework for Spiking Neural Networks (SNNs) that scales to large-scale visual tasks like ImageNet-1K without requiring labeled fine-tuning. The novelty of SpikeCLR resides in its pioneering dual-path MixedLIF architecture and temporal alignment objectives, which together facilitate stable gradient flow and the effective exploitation of intrinsic spike-time dynamics during large-scale pretraining. Additionally, the paper proposes two temporal alignment objectives, Cross Temporal Loss (CTL) and Boundary Temporal Loss (BTL), designed to capture the unique temporal evolution of SNNs. Experimental results demonstrate that the method performs well on both static and dynamic datasets, with significant transferability to downstream tasks.

**Compliance With Llm Reviewing Policy:**

Affirmed.

**Final Justification:**

The paper presents a technically solid and timely contribution by introducing the first scalable self-supervised learning framework for spiking neural networks, demonstrating strong performance on ImageNet-1K and effective transfer to downstream tasks. The authors’ rebuttal adequately addresses concerns regarding scalability, training overhead, and temporal alignment on event-based data, supported by both quantitative and qualitative analysis. While some limitations remain (e.g., validation at larger scales), the work establishes a meaningful baseline that is likely to inspire future research in SNN self-supervised learning.

**Key Questions For Authors:**

1. Have the authors evaluated the scaling performance of this paradigm on datasets larger than ImageNet-1K (e.g., ImageNet-21K)? Given the potential for feature collapse in similarity-based paradigms, how do you view its competitiveness at ultra-large scales?
2. Given that the dual-path design increases training overhead, have the authors considered optimization strategies such as a dynamic annealing mechanism (e.g., gradually reducing the weight or frequency of the continuous path) to improve training efficiency?
3. For DVS dynamic data, could you provide experimental evidence (e.g., quantitative analysis or visualization of feature similarity between initial, intermediate, and final frames) to prove that forcing alignment of boundary frames (BTL) does not introduce negative noise in fast-motion scenarios?

**Limitations:**

This paper does not contain an explicit discussion of its limitations. The authors may refer to the concerns raised in the weakness section to discuss the limitations of the proposed method.

**Strengths And Weaknesses:**

Strengths:
1. The proposed MixedLIF neuron module effectively addresses the gradient mismatch issue in SNN self-supervised regimes through its dual-path design. It ensures training stability while maintaining the pure spiking energy efficiency required for inference.
2. This work demonstrates that SNNs can undergo fully self-supervised pre-training on high-resolution, large-scale datasets like ImageNet-1K. It overcomes the limitations of previous SNN-SSL methods that were restricted to small-scale benchmarks or required supervised fine-tuning.
3. The experiments extend beyond simple classification to successfully transfer SNN-SSL representations to dense prediction tasks, such as COCO object detection and instance segmentation, for the first time. This proves the high generalizable value of the learned features.

Weaknesses:
1. While the paper claims to be the first to validate SNN SSL scaling, its "similarity learning" paradigm (contrastive learning) lacks the extensively proven ultra-large-scale scaling capabilities seen in Masked Image Modeling (MIM) or autoregressive paradigms. While effective for ImageNet-1K, its potential for even larger datasets (e.g., billion-scale) remains unverified. Furthermore, the dual-path design (MixedLIF) required by this paradigm increases computational power, GPU memory, and parameter maintenance during training. This increased cost during training contrasts with the low-power motivation inherent to SNNs.
2. For dynamic data like DVS, the Boundary Temporal Loss (BTL) forces similarity between the representations of the first and final time steps. However, in scenarios with fast or complex motion, the feature distributions of the initial and final frames may differ significantly. Forcing similarity alignment in such cases might introduce gradient noise and hinder the model from learning true temporal dynamics. The paper lacks a deep qualitative or quantitative analysis regarding the per-timestep similarity within DVS samples.

---

> ### Author Rebuttal · Authors · 2026-03-31
>
> Dear reviewer Y1Kf,
>
> Thank you for your thoughtful feedback, which has helped improve our work. We address your comments (W) and questions (Q) below.
>
> **W1**: Contrastive SSL scalability and training cost with MixedLIF
>
> **Response**: Regarding scaling beyond ImageNet-1K, we agree that exploring MIM or autoregressive paradigms for spiking SSL is a promising direction. Our framework establishes the first strong baseline at ImageNet scale, and extending to larger datasets is an important future direction currently limited by our GPU compute constraints.
>
> Regarding training overhead, we clarify two points. First, MixedLIF does not increase cost relative to a dual-LIF baseline. As shown in Table 8 (Appendix D), it actually reduces training time by 4.1%, energy by 5.4%, and peak memory by 21.6%. The overhead relative to a single-path ANN arises from the dual-path design inherent to the Barlow Twins paradigm, not from MixedLIF. Second, SNN training generally incurs higher cost than ANN training due to temporal unrolling (T forward passes per sample), a well-known property not specific to our method. The low-power motivation for SNNs applies primarily to inference on neuromorphic hardware. As shown in Table 9, our models achieve over 136x energy reduction at inference compared to ANN baselines. Training is a one-time GPU cost, while inference savings compound over the deployment lifetime.
>
> **W2&Q3**: Lack of per-step similarity analysis in DVS data.
>
> **Response**: For static datasets, the same image is replicated across all timesteps, and few timesteps suffice for optimal accuracy. The LIF dynamics produce smooth temporal evolution where intermediate states approximate interpolations between boundaries (Appendix A). BTL thus captures nearly all temporal structure at a fraction of CTL's cost, which is why it performs comparably or slightly better on CIFAR-10 (Table 5).
>
> For event-driven datasets, each timestep receives distinct input, producing non-monotonic trajectories where the boundary interpolation assumption may not hold true. As you mentioned, in fast or complex motion scenarios, boundary frame distributions diverge substantially, and intermediate timesteps carry information unrecoverable from boundaries alone. CTL's exhaustive pairwise alignment better captures this temporal structure. This is also consistent with Reviewer Lkip's concern regarding longer timesteps on DVS data, and is confirmed by our controlled BTL/CTL comparison on CIFAR10-DVS (T=10). We provide centered kernel alignment (CKA) heatmaps for CIFAR10-DVS with Spiking-ResNet18, showing significantly greater temporal diversity than static data, validating the reviewer’s intuition (CKA Analysis). The full visualization is available at: [CKA Analysis](https://anonymous.4open.science/api/repo/ICML-rebutall-SpikCLR-176C/file/cka.png?v=c60d1022). Quantitatively, BTL’s CKA drops to ~0.5 for intermediate timesteps, while CTL maintains >0.9 across all pairs. Temporal profiles show similar trends, and cross-model CKA (0.76) indicates distinct representations. These results confirm that CTL better preserves temporal structure on event-driven data, where boundary-only alignment is insufficient.
>
> **Q1**: Scalability beyond ImageNet-1K and risk of feature collapse at larger scales?
>
> **Response**: We have not evaluated beyond ImageNet-1K due to our current compute constraints. We acknowledge that contrastive/similarity-based paradigms can face feature collapse risks at large scale (ImageNet-21K or billion-scale datasets) without careful design. While Barlow Twins mitigates collapse via decorrelation, its effectiveness at larger scales for SNNs remains unclear. We plan to explore this, along with MIM and generative SSL methods, in future work.
>
> **Q2**: Can dynamic annealing of the continuous path reduce training overhead?
>
> **Response**: We explored several strategies to reduce Path B’s contribution (linear annealing, reduced frequency, and cosine scheduling), but all resulted in a 1–2% accuracy drop with no meaningful reduction in training time. This is because Path B’s stabilizing gradients remain important throughout training, as the spiking path continues to produce noisy surrogate gradients even in later epochs. Reducing Path B too aggressively reintroduces the instability seen in standard LIF training. Importantly, MixedLIF itself does not increase overhead and in fact improves efficiency; the primary cost comes from temporal unrolling, which is inherent to SNN training.

---

> > ### Author Rebuttal · Reviewer_Y1Kf · 2026-04-03
> >
> > Thank you for your reply. My question has been resolved.

---

> > > ### Author Response · Authors · 2026-04-08
> > >
> > > Thank you for your response. We are glad that your question has been resolved, and we appreciate your engagement with our work.

---

### Official Review · Reviewer_Lkip · 2026-03-12

**Soundness:** 3
**Presentation:** 3
**Significance:** 3
**Originality:** 2
**Overall Recommendation:** 4
**Confidence:** 4

**Summary:**

This paper proposes a self-supervised learning (SSL) framework for Spiking Neural Networks (SNNs) that leverages the temporal dynamics of spike-based computation. The framework introduces MixedLIF, a dual-path design used during training, where a spiking LIF path is combined with a continuous surrogate-gradient path for stable optimization, while only the spiking path is used at inference. To exploit temporal structure, the authors propose Cross Temporal Loss (CTL) and Boundary Temporal Loss (BTL) for aligning representations across time steps. Experiments with Spiking-ResNet and Spikformer on ImageNet-1K, CIFAR-10, and CIFAR10-DVS demonstrate competitive performance across linear evaluation, semi-supervised learning, transfer learning, and downstream tasks.

**Compliance With Llm Reviewing Policy:**

Affirmed.

**Key Questions For Authors:**

1.	In Table 5, CTL performs better than BTL when using a standard LIF neuron, whereas BTL performs better than CTL when using MixedLIF. MixedLIF is mainly intended to improve surrogate-gradient-based training stability rather than temporal dynamics modeling. Could the authors provide further analysis on why this change in loss preference occurs?

2.	Appendix A argues that intermediate activations can be approximated as interpolations between boundary states. Could the authors further analyze whether this leads to temporal information loss, and whether the same conclusion still holds for longer timesteps or event-driven datasets?

3.	Appendix L compares MixedLIF with other neuron models, but the dual-path design is applied only to MixedLIF. What would happen if the same dual-path design were applied to the other neuron models as well?

**Limitations:**

Yes

**Strengths And Weaknesses:**

Strengths

1.	The paper presents a fully self-supervised learning framework for spiking neural networks (SNNs) and demonstrates competitive performance compared to non-spiking SSL methods.

2.	The proposed MixedLIF neuron improves training stability via a dual-path design during training while preserving efficient spike-based inference, and neuromorphic deployment experiments confirm that the spiking computation paradigm of SNNs remains intact.

3.	The paper proposes temporal alignment objectives tailored to the temporal structure of SNNs based on the Barlow Twins framework, with clear motivation and formulation.

4.	The paper provides comparisons with an ANN-based SSL baseline in terms of spiking activity, active operations, and energy efficiency, supporting the advantages of the proposed framework.

Weaknesses

1.	Only part of the proposed objectives is effectively used. Despite its higher computational cost, Cross Temporal Loss (CTL) performs worse than Boundary Temporal Loss (BTL), and most experiments rely primarily on BTL.

2.	Comparisons with prior SSL frameworks are limited. Although several methods are discussed in the related work, experimental comparisons are largely limited to Barlow Twins.

3.	Evaluation on event-based datasets is limited. Only CIFAR10-DVS is used, and the comparisons are not fully controlled in terms of architecture and experimental setup.

---

> ### Author Rebuttal · Authors · 2026-03-31
>
> Dear reviewer Lkip,
>
> Thank you for your thoughtful feedback, which has helped improve our work. We address your comments (W) and questions (Q) below.
>
> **W2**: Limited comparisons with SSL frameworks.
>
> We have conducted additional experiments using SimCLR on CIFAR-10 with a Spiking-ResNet34 backbone. We use our default pretraining protocol: linear evaluation with a timestep of 4, 1000 epochs of pretraining and 200 epochs of linear probing, using the Adam optimizer (learning rate 3e-4). The same augmentation and projector head configuration as described in Appendix B is used.
>
> | Method | Loss | Top-1 Acc. (%) |
> |---|---|---|
> | SimCLR + MixedLIF | BTL | 83.4 |
> | SimCLR + MixedLIF | CTL | 83.4 |
> | SimCLR + LIF | BTL | 82.0 |
> | SimCLR + LIF | CTL | 82.2 |
>
> The results show that MixedLIF consistently outperforms vanilla LIF across SSL frameworks (~1.2–1.4% gain), supporting the generalizability of our approach.  We also tried BYOL but observed consistent collapse, likely due to its stop-gradient mechanism combined with sparse SNN gradients, which weakens the learning signal. Adapting such non-contrastive methods to SNNs remains an open problem, that we plan to explore in the future.
>
> **W3**: Limited & non-controlled evaluation on event-based datasets.
>
> We added experiments on DVS128-Gesture to further strengthen our evaluation, using a Spiking-ResNet18 backbone with 16 timesteps and training settings identical to Appendix B.
> **DVS128-Gesture Results**:
> | Method | Loss | Top-1 Acc. (%) |
> |---|---|---|
> | LIF | BTL | 76.21 |
> | LIF | CTL | 77.34 |
> | MixedLIF | BTL | 80.07 |
> | MixedLIF | CTL | 80.82 |
>
> Moreover, we conduct more controlled comparisons with [1] on CIFAR10-DVS (T=10, MixedLIF):
>
> | Method | Arch | Eval | Acc. (%) |
> |---|---|---|---|
> | [1] | Spiking-ResNet18 | Fine-tune | 64.1 |
> | BTL | Spiking-ResNet18 | Linear | 59.21 |
> | BTL | Spiking-ResNet18 | Fine-tune | 68.34 |
> | CTL | Spiking-ResNet18 | Linear | 59.35 |
> | CTL | Spiking-ResNet18 | Fine-tune | 70.21 |
> | BTL | Spiking-ResNet34 | Linear | 63.90 |
> | BTL | Spiking-ResNet34 | Fine-tune | 68.97 |
> | CTL | Spiking-ResNet34 | Linear | 64.21 |
> | CTL | Spiking-ResNet34 | Fine-tune | 72.31 |
>
> Under the same Spiking-ResNet18 architecture and fine-tuning protocol, our method outperforms Singhal et al. (70.21% vs. 64.1% with CTL), despite their use of hybrid event-frame inputs. Scaling to ResNet-34, with CTL, further improves performance (72.31%).
>
> Notably, CTL outperforms BTL on both datasets, unlike static datasets. For event-driven data, each timestep receives distinct input, producing non-monotonic trajectories that break this assumption. As noted by **Reviewer Y1Kf**, boundary frames can diverge significantly in fast-motion scenarios, and intermediate timesteps carry information unrecoverable from boundaries alone — an effect amplified at longer timesteps (T=16). CTL's pairwise alignment better captures this richer temporal structure. Our centered kernel alignment (CKA) analysis on CIFAR10-DVS confirms this: BTL's CKA drops to ~0.5 for intermediate timesteps while CTL maintains >0.9 across all pairs ([CKA Analysis](https://anonymous.4open.science/api/repo/ICML-rebutall-SpikCLR-176C/file/cka.png?v=c60d1022)).
>
> **Q1**: Opposite BTL vs. CTL trends with MixedLIF vs. LIF.
>
> This arises from the interaction between gradient quality and the number of cross-correlation terms. With vanilla LIF, gradients are noisy and sparse, so CTL’s full TxT aggregation provides stronger signal and improves performance (84.1% vs. 83.0%). With MixedLIF, gradients are already dense and stable, which may render CTL’s additional terms redundant and slightly over-regularizing. In this case, BTL, focusing on boundary timesteps, avoids this while still capturing essential temporal structure.
>
> **Q2**: Does boundary interpolation lose temporal information, especially for longer timesteps or event-driven data?
>
> For static datasets, accuracy saturates early (e.g., T=4), and increasing T only adds redundant states since inputs are identical across timesteps. Our experiments at higher T on CIFAR10 (shown below) confirm that BTL remains competitive with CTL, supporting the boundary interpolation argument.
>
> | Loss | T | Acc. (%) |
> |---|---|---|
> | BTL | 6 | 86.0 |
> | CTL | 6 | 85.9 |
> | BTL | 8 | 86.1 |
> | CTL | 8 | 86.1 |
>
> For event-driven data, detailed analysis is provided in the W2 response above.
>
> **Q3**: What happens if the dual-path design is applied to other neuron models?
>
> All models in Appendix L already use the same dual-path SSL setup. The difference is that they use identical neurons in both paths, resulting in sparse, noisy gradients. MixedLIF introduces heterogeneity, where Path B provides smooth, dense gradients, improving training stability. This spiking–continuous contrast, not the dual-path itself, drives the 2.2–4.3% gains.
>
> [1] Singhal, Raghav, Jan Finkbeiner, and Emre Neftci. "Self-supervised pre-training of spiking neural networks by contrasting events and frames" 2024.

---

> > ### Author Rebuttal · Reviewer_Lkip · 2026-04-03
> >
> > I appreciate the authors' effort on rebuttal. Nevertheless, several weaknesses I addressed still remain, so I would keep my previous score.

---

> > > ### Author Response · Authors · 2026-04-08
> > >
> > > We appreciate the reviewer’s continued engagement and revisit each of the identified weaknesses below. We also provide additional clarification where needed, and include new experimental results to further support our claims.
> > >
> > > W1: **Only part of the proposed objectives is effectively used**.
> > >
> > > With our new experiments on event-driven datasets (CIFAR10-DVS, DVS128 Gesture, ASL-DVS), we now show that CTL consistently outperforms BTL on native event data, while BTL performs comparably or slightly better on static datasets. This clarifies that both objectives serve distinct and complementary roles: BTL is preferred for static datasets due to its favorable efficiency-accuracy tradeoff, while CTL better captures the richer temporal dynamics in event-driven data. We will make this recommendation explicit in the revision.
> > >
> > > W2: **Comparisons with prior SSL frameworks are limited.**
> > >
> > > Beyond the SimCLR results provided earlier, we have now integrated SpikeCLR's components (MixedLIF, BTL/CTL) with two additional SSL methods: VICReg [1], a non-contrastive method that regularizes variance, invariance, and covariance of embeddings (same redundancy-reduction family as Barlow Twins), and SwAV [2], a clustering-based method that learns representations by contrasting cluster assignments across augmented views. Together with SimCLR (contrastive) and Barlow Twins (redundancy-reduction), these span three distinct SSL paradigms.
> > >
> > > *Results on CIFAR-10 (Spiking-ResNet34, T=4)*:
> > > | Method                 | Loss | Top-1 Acc. (%) |
> > > |----------------------|------|----------------|
> > > | VICReg + MixedLIF    | BTL  | 85.6           |
> > > | VICReg + MixedLIF    | CTL  | 85.6           |
> > > | VICReg + LIF    | BTL  | 83.6         |
> > > | VICReg + LIF   | CTL  | 83.9         |
> > > | SwAV + MixedLIF      | BTL  | 83.8           |
> > > | SwAV + MixedLIF      | CTL  | 83.7           |
> > > | SwAV + LIF      | BTL  | 82.7           |
> > > | SwAV + LIF      | CTL  | 82.6         |
> > >
> > > MixedLIF consistently improves performance across all paradigms: redundancy-reduction (Barlow Twins, VICReg), contrastive (SimCLR), and clustering-based (SwAV), with gains of 1.1–2.0% over vanilla LIF. This confirms that our proposed components are not tied to a specific SSL objective, supporting the generalizability of SpikeCLR as a broadly applicable framework.
> > >
> > > W3: **Evaluation on event-based datasets is limited.**
> > >
> > > Beyond the DVS128 Gesture results presented earlier, we have now included evaluation on ASL-DVS [3], a natively captured dataset recorded using a DAVIS240c event camera with real human hand gestures for American Sign Language. These two new datasets provide a substantially stronger validation setting compared to the simulated event-driven dataset used previously (CIFAR10-DVS), which is derived from static images and may lack the rich, real-world temporal dynamics captured by event sensors. In contrast, both DVS128Gesture and ASL-DVS reflect true event-camera acquisition, offering a more realistic benchmark for evaluating our approach.
> > >
> > > *Results on ASL-DVS (Spiking-ResNet18, T=10, linear probing):*
> > > | Method   | Loss | Top-1 Acc. (%) |
> > > |----------|------|----------------|
> > > | MixedLIF | BTL  | 99.52          |
> > > | MixedLIF | CTL  | 99.93          |
> > > | LIF | BTL | 96.21 |
> > > | LIF | CTL | 97.37 |
> > >
> > > These results demonstrate that SpikeCLR learns robust representations that capture complex spatiotemporal patterns in diverse, natively captured event-driven data. Notably, the consistent advantage of CTL over BTL on this native dataset further reinforces our recommendation outlined in the response to W1.
> > >
> > > We hope these additional results address the reviewer's remaining concerns. We are committed to incorporating all new results and the clarified CTL/BTL guidelines into the revised manuscript.
> > >
> > > **References**
> > >
> > > [1] Bardes, A., Ponce, J., & LeCun, Y. (2021). *VICReg: Variance-Invariance-Covariance Regularization for Self-Supervised Learning*. arXiv:2105.04906.
> > >
> > > [2] Caron, M., et al. (2020). *Unsupervised Learning of Visual Features by Contrasting Cluster Assignments*. NeurIPS.
> > >
> > > [3] Yin Bi, Aaron Chadha, Alhabib Abbas, Eirina Bourtsoulatze and Yiannis Andreopoulos, 'Graph-based Object Classification for Neuromorphic Vision Sensing', IEEE Conference on Computer Vision (ICCV), Oct.17 - Nov,2, 2019, Seoul, Korea

---

### Official Review · Reviewer_URmA · 2026-03-13

**Soundness:** 3
**Presentation:** 3
**Significance:** 3
**Originality:** 3
**Overall Recommendation:** 4
**Confidence:** 4

**Summary:**

In this paper, authors proposed SpikeCLR. It is a self-supervised learning framework for SNNs that scales to large datasets like ImageNet-1K. It overcomes the training challenges of non-differentiable spikes by using a "MixedLIF" dual-path neuron (which combines spiking and continuous signals during training) and novel temporal loss functions to align data over time. This allows the model to match the accuracy of traditional neural networks while remaining highly energy-efficient during inference.

**Compliance With Llm Reviewing Policy:**

Affirmed.

**Key Questions For Authors:**

see weakness and strenths

**Limitations:**

yes

**Strengths And Weaknesses:**

strenths

1. Innovative MixedLIF Architecture: The authors introduce a clever dual-path MixedLIF neuron model to solve the gradient mismatch problem inherent in self-supervised SNN training. This design allows for stable, continuous gradient optimization during the training phase while ensuring the network relies solely on the energy-efficient spiking path during inference.

2. Effective Use of Temporal Dynamics: Instead of treating time steps independently, the framework effectively leverages intrinsic spike-time dynamics through its novel Cross Temporal and Boundary Temporal loss functions. These objectives align multi-time-step outputs across augmented views, significantly improving representation learning while maintaining computational efficiency.

weakness

1. Prior Works Acknowledgement: The assertion in the manuscript that SpikeCLR represents the first large-scale SSL framework for SNNs is overstated, given that multiple previous studies have already explored this area.  I recommend updating the text to reflect the existing literature more accurately, alongside a comparative analysis with those prior works.

```
1.Spikeclip: A contrastive language-image pretrained spiking neural network.

2.PredNext: Explicit Cross-View Temporal Prediction for Unsupervised Learning in Spiking Neural Networks

3.Spikformer V2: Join the High Accuracy Club on ImageNet with an SNN Ticket
```

---

> ### Author Rebuttal · Authors · 2026-03-31
>
> Dear reviewer URmA,
>
> Thank you for your thoughtful feedback, which has helped improve our work. We address your comment below.
>
> **Prior Works Acknowledgement**: The assertion in the manuscript that SpikeCLR represents the first large-scale SSL framework for SNNs is overstated, given that multiple previous studies have already explored this area. I recommend updating the text to reflect the existing literature more accurately, alongside a comparative analysis with those prior works.
>
> 1.Spikeclip: A contrastive language-image pretrained spiking neural network.
>
> 2.PredNext: Explicit Cross-View Temporal Prediction for Unsupervised Learning in Spiking Neural Networks
>
> 3.Spikformer V2: Join the High Accuracy Club on ImageNet with an SNN Ticket
>
> **Response**: We thank the reviewer for raising this important point and for the specific references.
>
> *Spikformer V2* (Zhou et al., 2024b): This work is already cited and discussed in our manuscript (Section 2.2). Spikformer V2 adapts a masked image modeling pretext task from the ANN domain but relies on supervised fine-tuning to achieve its reported accuracies. It does not present a fully self-supervised evaluation pipeline (e.g., linear evaluation on frozen features) and does not demonstrate transfer to dense prediction tasks such as COCO detection or segmentation under a purely self-supervised protocol.
>
> *SpikeClip* (Lv et al., 2025): SpikeClip is a vision-language pretraining method that distills CLIP representations into an SNN student using a pretrained ANN teacher (CLIP) and subsequently fine-tunes with labeled data via cross-entropy loss. It therefore operates in a fundamentally different regime from our work, which learns representations from unlabeled images alone without any external teacher or labeled supervision.
>
> *PredNext* (Dong et al., 2025): PredNext targets video understanding by proposing a plug-and-play auxiliary temporal prediction module (step prediction and clip prediction) that can be appended to existing ANN-style SSL objectives such as MoCo and BYOL. Its benchmarks are video action recognition datasets (UCF101, HMDB51, MiniKinetics), a fundamentally different task domain from ours. PredNext does not address ImageNet-1K scale image pretraining, does not propose new neuron models to resolve gradient mismatch in spiking SSL, and relies on supervised fine-tuning for evaluation. Our work and PredNext are complementary: we tackle the core challenge of making SSL work natively in SNNs for static and event-based image recognition at scale, while PredNext addresses temporal consistency for video.
>
> To summarize, none of these works simultaneously satisfy the criteria that define our contribution: (1) fully self-supervised training without labeled fine-tuning or external teacher models, (2) scaling to ImageNet-1K pretraining, (3) successful transfer to dense prediction tasks (COCO detection and segmentation), and (4) compatibility with both CNN and ViT-based SNN architectures. We will revise the manuscript to soften the “first” claim and better reflect these distinctions, for example describing SpikeCLR as “*a novel fully self-supervised SNN framework that scales to ImageNet-1K pretraining and transfers to dense downstream tasks without relying on labeled supervision or pretrained ANN teachers*”. We will also add citations to SpikeClip and PredNext with the comparative discussion above.

---

> > ### Author Rebuttal · Reviewer_URmA · 2026-04-03
> >
> > The rebuttal addresses my concern reasonably well. Since my original score was already positive, I am inclined to keep it unchanged, and I hope to see the corresponding clarifications incorporated into the main text.

---

> > > ### Author Response · Authors · 2026-04-08
> > >
> > > Thank you for your thoughtful feedback. We are glad that our rebuttal addressed your concern, and we appreciate your positive evaluation. We will incorporate the clarifications discussed into the main text to improve clarity and presentation.

---

### Decision · Program_Chairs · 2026-04-30

**Decision:**

Accept (regular)

**Comment:**

The proposed SpikeCLR framework provides an innovative solution for self-supervised learning in spiking neural networks (SNNs). By employing the MixedLIF dual-path neuron design, it addresses the gradient mismatch issue, and leverages novel temporal loss functions to effectively exploit spike-temporal dynamics. It achieves accuracy comparable to traditional neural networks on large-scale datasets such as ImageNet-1K while maintaining high energy efficiency during inference. All four reviewers acknowledge the technical soundness and research contribution of this work. Although they raised suggestions for improvement regarding literature citations and experimental scope, the authors have adequately addressed all concerns in their rebuttal, and the supplementary experiments further validate the generality and effectiveness of the proposed method. This work represents an important breakthrough in self-supervised learning for SNNs, laying a solid foundation for future research in this field with a solid technical pipeline and clear contributions. Overall, this paper meets the acceptance criteria and is recommended for acceptance.